# Periodic synchronisation of dengue epidemics in Thailand over the last 5 decades driven by temperature and immunity

**Bernardo García-Carreras** [1,2]*, **Bingyi Yang** [1,2¤], **Mary K. Grabowski**[3], **Lawrence W. Sheppard**[4,5], **Angkana T. Huang** [1,2,6], **Henrik Salje**[7], **Hannah Eleanor Clapham**[8], **Sopon Iamsirithaworn**[9], **Pawinee Doung-Ngern**[9], **Justin Lessler** [10], **Derek A. T. Cummings** [1,2]

**1** Department of Biology, University of Florida, Gainesville, Florida, United States of America, **2** Emerging Pathogens Institute, University of Florida, Gainesville, Florida, United States of America, **3** Department of Pathology, Johns Hopkins School of Medicine, Baltimore, Maryland, United States of America, **4** Department of Ecology and Evolutionary Biology and Kansas Biological Survey, University of Kansas, Lawrence, Kansas, United States of America, **5** The Marine Biological Association, Plymouth, United Kingdom, **6** Department of Virology, Armed Forces Research Institute of Medical Sciences, Bangkok, Thailand, **7** Department of Genetics, University of Cambridge, Cambridge, United Kingdom, **8** Saw Swee Hock School of Public Health, National University of Singapore, Singapore, Singapore, **9** Department of Disease Control, Ministry of Public Health, Nonthaburi, Thailand, **10** Department of Epidemiology, Johns Hopkins Bloomberg School of Public Health, Baltimore, Maryland, United States of America

¤ Current address: WHO Collaborating Centre for Infectious Disease Epidemiology and Control, School of Public Health, Li Ka Shing Faculty of Medicine, The University of Hong Kong, Hong Kong Special Administrative Region, China

* bgarciacarreras@gmail.com

**Data Availability Statement:** The data presented in every figure is available at https://github.com/UF-IDD/synchrony_dengue_figures.

## Abstract

The spatial distribution of dengue and its vectors (spp. *Aedes*) may be the widest it has ever been, and projections suggest that climate change may allow the expansion to continue. However, less work has been done to understand how climate variability and change affects dengue in regions where the pathogen is already endemic. In these areas, the waxing and waning of immunity has a large impact on temporal dynamics of cases of dengue haemorrhagic fever. Here, we use 51 years of data across 72 provinces and characterise spatiotemporal patterns of dengue in Thailand, where dengue has caused almost 1.5 million cases over the last 30 years, and examine the roles played by temperature and dynamics of immunity in giving rise to those patterns. We find that timescales of multiannual oscillations in dengue vary in space and time and uncover an interesting spatial phenomenon: Thailand has experienced multiple, periodic synchronisation events. We show that although patterns in synchrony of dengue are similar to those observed in temperature, the relationship between the two is most consistent during synchronous periods, while during asynchronous periods, temperature plays a less prominent role. With simulations from temperature-driven models, we explore how dynamics of immunity interact with temperature to produce the observed patterns in synchrony. The simulations produced patterns in synchrony that were similar to observations, supporting an important role of immunity. We demonstrate that multiannual oscillations produced by immunity can lead to asynchronous dynamics and that synchrony in temperature can then synchronise these dengue dynamics. At higher mean

**Funding:** BGC, BY, ATH, HS, and DATC were funded by NIH National Institute of Allergy and Infectious Diseases grant R01 AI114703-01. The funders had no role in study design, data collection and analysis, decision to publish, or preparation of the manuscript.

**Competing interests:** The authors have declared that no competing interests exist.

**Abbreviations:** CWMF, cross-wavelet mean field; DENV, dengue virus; DF, dengue fever; DHF, dengue haemorrhagic fever; DSS, dengue shock syndrome; ENSO, El Niño Southern Oscillation; GHCN CAMS, Global Historical Climatology Network version 2 and the Climate Anomaly Monitoring System; WMF, wavelet mean field; WPMF, wavelet phasor mean field; WT, wavelet transform.

temperatures, immune dynamics can be more predominant, and dengue dynamics more insensitive to multiannual fluctuations in temperature, suggesting that with rising mean temperatures, dengue dynamics may become increasingly asynchronous. These findings can help underpin predictions of disease patterns as global temperatures rise.

## Introduction

The spatiotemporal dynamics of animal populations, including fluctuations and correlations in amplitudes and phases, has been an important area of research in ecology and the physical sciences for decades. Empirical observations of patterns in population dynamics have propelled theory forward leading to a better understanding of predator–prey interactions [1,2], allee effects [3], and the interactions between deterministic and stochastic elements of systems [4,5]. While a rich literature exists on synchronisation of systems and their causes [6], there is little empirical evidence for periodic oscillations or fluctuations in spatial synchrony, particularly in epidemiology, where identification of long-term patterns in infectious disease dynamics could be critical for the success of health interventions. For instance, knowing when pathogen population levels are particularly low concurrently across a region presents improved opportunities for pathogen elimination [7]. Likewise, anticipation of global epidemics may assist in the structured allocation of resources, such as vaccines, across space and time. While regular spatial synchrony in other wildlife species has been described (e.g., [8–12]), how the degree of synchrony may vary over time has received less attention [13–15]. Here, we describe regular periodic synchronisation in the dynamics of dengue haemorrhagic fever (DHF) in Thailand over a 51-year interval.

Dengue virus (DENV) is a mosquito-borne virus estimated to infect 100 million people each year [16,17]. Four viral serotypes (DENV1–4) exist. Primary infection with a specific serotype confers long-term immunity against subsequent infections of that same serotype, and there is strong empirical support for short-term, temporary protection against other serotypes [18–21]. Additionally, other mechanisms such as antibody-dependent enhancement also potentially mediate the dynamics of incidence [22–24].

Multiannual patterns in dengue observed in any location arise as the result of a complex interplay of various factors, including the following: climate, through its effect on the vector and transmission efficiency; predator–prey dynamics between the virus and the host (characterised by phase-shifted fluctuations in the predator and prey abundances); the interactions between different serotypes and strains of dengue; spatial patterns in host structure, dynamics, and movement; and viral factors [25–28]. Much work has gone into disentangling the roles of extrinsic drivers, such as climate, and intrinsic factors in shaping dynamics across many disease systems and using a range of approaches [29–34]. In ascribing drivers to interannual patterns in dengue dynamics, studies have often viewed extrinsic factors and intrinsic factors, particularly the dynamics of immunity, as competing alternative hypotheses [26,35–41]. However, immunity clearly provides a negative feedback where increases in transmission due to favourable climatic conditions can lead to decreases in transmission in future time periods through the protective effects of immunity, leading to multiannual dynamics in many systems [30,42].

Many of the same factors involved in generating multiannual dynamics can also produce synchrony across locations [6]. Again, both intrinsic (specifically host movement between locations) and extrinsic (environmental; the "Moran effect," where correlations between

geographically separate populations are driven by correlations in their respective environments) have the potential to not only synchronise dynamics across space, but also produce variation in the degree of synchrony over time [13–15]. Empirical studies have tended to focus on climate [40,41]. van Panhuis and colleagues [40], for instance, detected synchronous dengue outbreaks across Southeast Asia in 1997 to 1998, a period of elevated temperatures and a strong El Niño event.

The environment, and especially temperature, is a priori expected to be significant in shaping the dynamics of dengue [43–45]. Through its effect on metabolic rates, temperature strongly influences many mosquito life history traits (e.g., biting rates, population growth rates, mortality rates; [28,44]) and, therefore, their population dynamics [46]. Projections suggest that increasing temperatures may expand the range in which dengue is efficiently transmitted [47–49]. However, the impact of changing temperature regimes on dengue in locations where the pathogen is already endemic is less well studied, and these areas, even in projections to 2050 or 2080, comprise a majority of the world's population at risk of dengue [47–49].

Here, we examine the spatiotemporal dynamics of dengue in Thailand between 1968 and 2018. We characterise multiannual cycles and uncover an interesting spatial phenomenon in which Thailand has experienced periodic synchronisations of dengue incidence. In contrast to previous studies, which tended to focus on either extrinsic or intrinsic drivers to explain observed patterns, we hypothesise that immunity constitutes a strong dynamical filter necessary to understand impacts of temperature on dengue. For this reason, we adapt a mechanistic, temperature-dependent, 4-serotype dengue model to disentangle how temperature and dynamics of immunity interact to generate periodic synchronisations. We focus on temporary cross-protection between serotypes because of the clear empirical support, although there are other mechanisms, such as antibody-dependent enhancement, that could conceivably also play a role.

## Results

### Empirical patterns in multiannual cycles and synchrony

**Multiannual cycles in dengue.** Fig 1A shows a heatmap of the monthly number of dengue cases by province over the 51 years, in which seasonal outbreaks are clear, but in which some years (e.g., 1998 to 1999 or 2001) had larger outbreaks throughout the country than other years. It is these latter multiannual patterns that we are interested in here. Specifically, we focus on cyclical patterns with timescales between 1.5 and 5 years (i.e., patterns in which years with larger than average outbreaks are separated by 1.5 to 5 years) because this range encompasses the timescales that have been previously reported for dengue dynamics (see section "Materials and methods" for a more detailed rationale). To characterise multiannual cycles, we applied continuous wavelet transforms (WTs). WTs quantify the importance of different timescales in a time series over time, and with them, time series can be reconstructed only using the multiannual components of observed patterns. Reconstructions of the dengue time series confirm the presence of distinct multiannual patterns (Fig 1B). We quantify which multiannual timescale is most important in characterising dengue dynamics in each province and for each point in time (the "dominant timescale"), revealing both that the dominant multiannual timescale tends to change over time (varying between 1.5 and 4.5 years) and that at any given point in time, the dominant timescales can differ substantially across provinces (S8b Fig in S1 Appendix). The presence of multiannual timescales has been previously observed [27,40], but here, with the benefit of longer time series, we show that these change over time and space.

**Synchrony in dengue.** We estimate spatial synchrony in dengue cases using a range of methods; we here focus on one approach, wavelet mean fields (WMFs; [50,51]), but provide

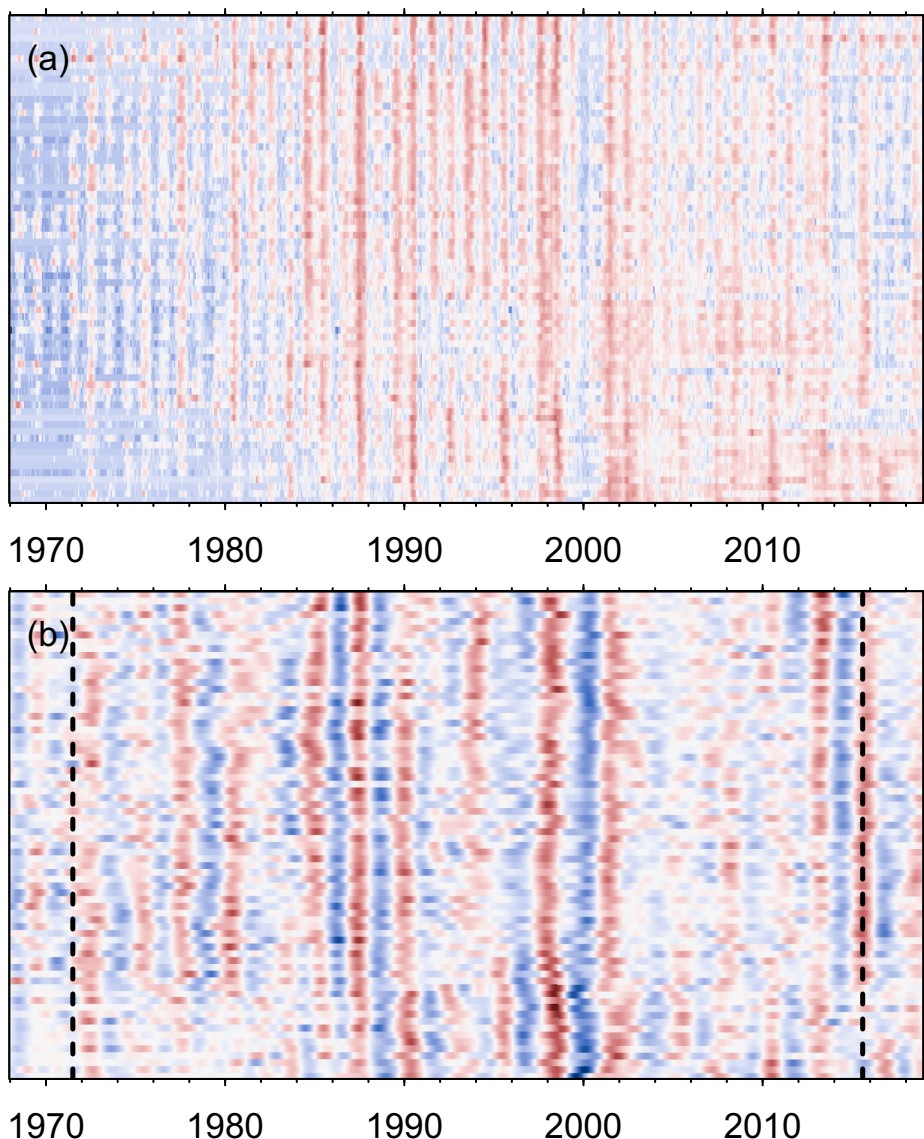

**Fig 1. Dengue multiannual cycles.** Heatmaps of (a) ln number of cases and (b) reconstructions of time series using multiannual components only, per province arranged from north (top) to south (bottom). To improve clarity, values for each province were normalised to a mean of 0 and a standard deviation of 1 in both panels. Blues (respectively reds) are lower (respectively higher) numbers, and whites correspond to the mean for each province. Edge effects in the WTs may influence results before and after the vertical dashed lines in (b) (see section "Materials and methods"). The underlying data are in S1 Data at https://github.com/UF-IDD/synchrony_dengue_figures. WT, wavelet transform.

details and results for the others in section "Perspectives on synchrony" in S1 Appendix. WMFs measure synchrony as a function of both timescale and time; they indicate timescales and time points at which both phases (the position in time on a cycle) and magnitude of oscillations are consistent (or more synchronous) across provinces. The WMF for DHF cases across Thailand describes a system that appears to fluctuate in and out of synchrony (Fig 2B), results that are consistent with the travelling waves observed across Thailand [27] and Southeast Asia [40]. The synchrony in dengue is statistically significant at all times, meaning that even when synchrony is lower (the whiter areas in Fig 2B), the degree of synchrony is still

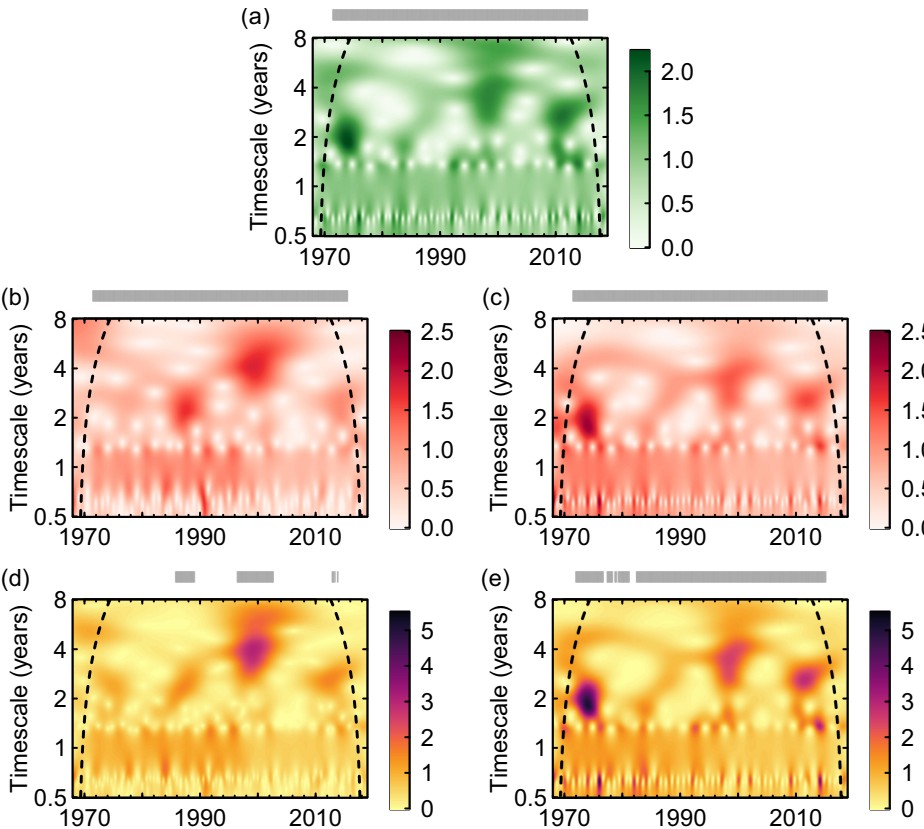

**Fig 2. (Cross-)WMFs.** WMFs for (a) temperature, and ln-transformed dengue cases from the (b) passive surveillance data and (c) model output. Cross-WMFs between (d) temperature and ln-transformed dengue cases (data) and (e) temperature and ln-transformed dengue model output. Model output assumes a mean cross-protection of 1 year (see S13 Fig in S1 Appendix for results using other mean cross-protections). The same temperature time series are used for both (d, e). Grey bars above panels indicate the times for which phases (in a–c) and phase difference angles (in d, e) are highly consistent across provinces and statistically significant (see section "Material and methods"). In (a–c), higher values in the mean fields indicate timescales and points in time where the phases are more consistent across provinces, and where the amplitudes of oscillations are more correlated. In (d, e), higher values correspond to timescales and points in time where the agreement between dengue and temperature is itself more consistent across provinces. S14 Fig in S1 Appendix shows which multiannual timescales dominate the (C) WMFs for each panel in this figure. Edge effects in the WTs may influence results before and after the dashed lines (see section "Materials and methods"). The underlying data are in S1 Data at https://github.com/UF-IDD/synchrony_dengue_figures. WMF, wavelet mean field; WT, wavelet transform.

greater than would be expected in an asynchronous system. Furthermore, moments of greater synchrony take place at different timescales. For example, while the synchrony event of the late 1980s occurs with a timescale of ≈2.2 years, the one taking place around the year 2000 has a timescale of ≈4.1 years. These results are further corroborated using 4 additional approaches to estimate synchrony (Fig 3); all methods describe a system that appears to fluctuate in and out of synchrony (section "Perspectives on synchrony" in S1 Appendix). In all cases, periods of greater synchrony appear to coincide with larger outbreaks (e.g., comparing S8a and S8c Fig in S1 Appendix), as also previously observed [40,52].

**Synchrony in temperature.** Using the Global Historical Climatology Network version 2 and the Climate Anomaly Monitoring System (GHCN CAMS), we extracted a temperature time series for each province and calculated the temperature WMF across provinces for the

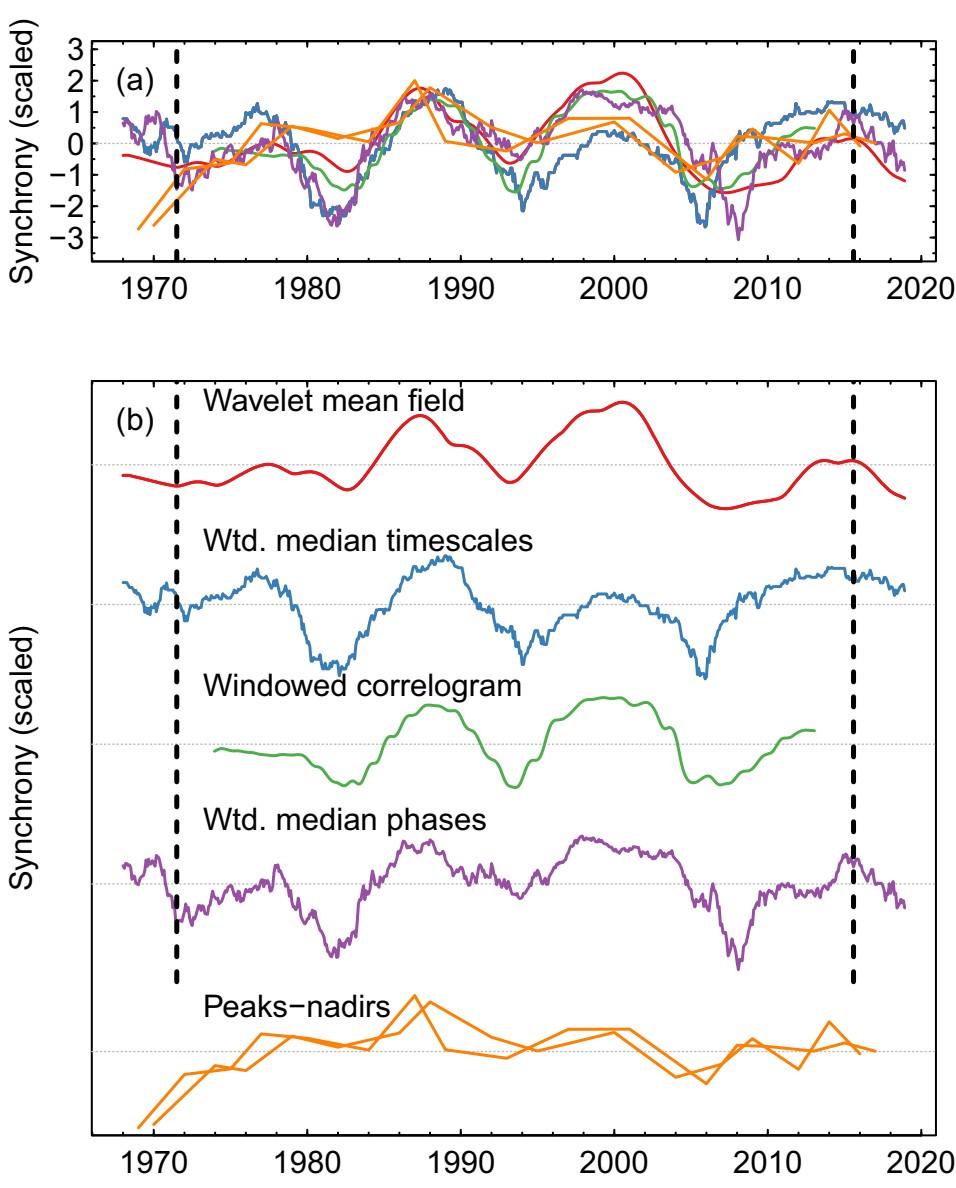

**Fig 3. Comparison of all measures of synchrony in dengue.** For details on these measures of synchrony, see section "Perspectives on synchrony" in S1 Appendix. The curve for WMFs was obtained by taking the mean WMF of Fig 2B across multiannual timescales, for each point in time. To facilitate comparisons, all metrics were normalised to have a mean of 0 and standard deviation of 1, and where necessary, time series were flipped along the y-axis so that higher values always equate to greater synchrony. Note that for the metric taken from the distribution of distances of peaks and nadirs, there are 2 time series in the same colour (one for peaks and another for nadirs). The 2 panels show the same time series, but (b) separates them for clarity. Edge effects in the WTs may influence results before and after the vertical dashed lines. The underlying data are in S1 Data at https://github.com/UF-IDD/synchrony_dengue_figures. WMF, wavelet mean field; WT, wavelet transform.

same time period as the dengue passive surveillance data (1968 to 2018). As expected, synchrony in temperature was significant at all points in time. The WMF for temperature describes a system similar to that found in dengue, where temperature fluctuates in and out of synchrony at multiannual timescales (Fig 2A; for results using alternative approaches, see S34

and S35 Figs in S1 Appendix). Furthermore, the moments of greater synchrony appear to align, both in timing and timescale, with those of dengue (Fig 2A and 2B).

If the patterns in synchrony in dengue were to be directly linked to patterns in synchrony in temperature, we would expect the magnitudes of the 2 respective WMFs (in the multiannual time scales, and for all points in time excluding the cone of influence) to correlate positively and significantly. We found a Pearson correlation of 0.21 ($P = 0.060$; see section "Materials and methods" for significance testing) and a Spearman correlation of 0.20 ($P = 0.069$). Thus, while correlations were positive, they were not statistically significant. However, see the result for WMFs of dengue data and the output of a mechanistic model based on temperature in section "Driving the model with actual temperature time series" below.

To address whether the patterns of synchrony in temperature were consistent with those observed in the dengue cases in greater detail, we extended the methods of Sheppard and colleagues [50,51] to calculate cross-wavelet mean fields (CWMFs). Cross-wavelets quantify the similarity in 2 time series' (temperature and dengue cases) wavelet power and phase angles at each timescale and time point, so the CWMFs (the weighted mean cross-wavelet across provinces) quantify the consistency of this similarity across locations. Results in Fig 2D (in which darker colours mean greater consistency between dengue and temperature across locations) suggest that temperature plays a crucial role particularly when synchrony in dengue was high (as demonstrated by the statistically significant consistency in phase angles between dengue and temperature during synchrony events).

On the other hand, the CWMFs do not clarify what mechanisms are at play during the more asynchronous periods. Asynchronous multiannual dynamics in temperature might lead to similar asynchronous multiannual dynamics in dengue, or different thermal regimes might elicit intrinsic multiannual dynamics of varying periodicities in dengue, thus also leading to asynchrony. In both these cases, the CWMF might be expected to be low. To distinguish between these 2 hypotheses, for each province, we correlated the wavelet power in temperature and dengue, separately for high- and low-synchrony periods (S15 Fig in S1 Appendix). Having established the importance of temperature during synchronous events, we expected statistically significant positive correlation coefficients during high-synchrony periods. Were temperature to also have been the main driver during the asynchronous periods, we might have expected similar correlation coefficients. However, we found correlations to be significantly lower during low synchrony (S16 Fig and S2 Table in S1 Appendix). This result alone does not preclude temperature from being the main driver of dengue dynamics during asynchronous periods, but see section "Driving the model with actual temperature time series."

**Periodicity of synchronisation events in dengue and temperature.** Our analyses describe a system that oscillated in and out of synchrony (Figs 2B and 3). To describe the periodicity of the degree of synchrony itself, we collapsed WMFs to a time series by taking the mean WMF across multiannual timescales for each point in time (red line in Fig 3B). We then normalised the resulting time series to a mean of 0 and standard deviation of 1 and applied WTs.

Because these time series of synchrony contain, at most, 4 distinct cycles (Fig 3), we chose to characterise their periodicity by averaging the wavelet power over time, by taking the mean wavelet power per timescale. The periodicities that dominated the time series of synchrony of dengue and temperature in Thailand, where synchrony was quantified using WMFs, were 12.6 and 12.5 years, respectively (e.g., the main periodicity of the cycles shown in the red line in Fig 3 is 12.6 years). As a point of comparison, the periodicity of synchrony in dengue, where synchrony is estimated using alternative methods such as weighted median timescales or windowed spline correlograms (section "Perspectives on synchrony" in S1 Appendix) was 12.0 years in both cases.

## Simulations to explore mechanism: The interplay of temperature and immunity

**Mechanistic dengue model and simulations.** While our results may suggest a relatively straightforward statistical relationship between dengue cases and temperature (Fig 2 and S36 Fig in S1 Appendix), we wanted to explore the mechanisms by which temperature could generate asynchronous and synchronous dynamics, while accounting for intrinsic factors, such as the interaction between serotypes. Immunity is central in shaping dengue dynamics; temporary cross-protection between serotypes alone can give rise to a qualitatively wide range of dengue dynamics. Any effects of temperature on dynamics must therefore necessarily be viewed through the lens of the dynamics of immunity. To this end, we used a simple, temperature-dependent 4-serotype differential equation dengue model (see section "Materials and methods"). The model, building on that of Huber and colleagues [46], explicitly encodes the temperature dependence of vector traits, which, in turn, allows for temperature to drive transmission and dengue dynamics. We extended their model to allow for cross-protective interactions between serotypes of varying mean lengths of time.

The patterns quantified so far describe a system that fluctuates in and out of synchrony. When seeking potential drivers for the observed patterns, these would need to account for both asynchrony, and thus the ability to produce a range of spatially heterogeneous dynamics, and synchrony, during which dynamics are more homogeneous.

We ran 3 sets of simulations to see whether we could better understand how temperature might drive patterns in synchrony, using both real temperature time series and simplified experiments (Table 1). First, in simulation 1, we used the temperature time series for each province as input for the model, with the objective of reproducing observed spatiotemporal features. Next, for simulations 2 and 3, we wanted to delve into the specific mechanisms through which temperature may produce asynchronous and synchronous dynamics. Simulation 1 allows us to establish that temperature plays a less important role in producing asynchronous dynamics in dengue; we therefore explore whether different temperature (and thus transmission) regimes can generate diverse enough (and thus asynchronous) intrinsic multiannual oscillations across locations. To this end, we devised simplified experiments across 6 idealised hypothetical locations along a transect in Thailand, going from high mean temperatures and low seasonal variability, to low mean temperatures and high seasonal variability (see S40–S42 Figs in S1 Appendix). For simulation 2, we used seasonal sine curves as temperature inputs (with no multiannual cycles) and ran simulations for the 6 locations. Finally, for simulation 3, we built on simulation 2 by introducing a single multiannual fluctuation in the temperature time series, at the same time across the 6 locations, superimposed over the seasonal cycles. The multiannual fluctuation had varying timescales (2 to 5 years), and the amplitude of

**Table 1. Simulations used in this study.** We use the mechanistic dengue model, with different temperature inputs, numbers of locations $L$, and initial population sizes per location $N$, to address specific but related questions (column Aim). Each location was run as an independent simulation (no host movement between locations), with a total population size corresponding to the population size of each province (simulation 1), or equal across locations (simulations 2 and 3), using the same host birth and death rates, such that the only differences across locations were temperature (all simulations), and population size (simulation 1 only; see section "Further details on simulation studies" in S1 Appendix). In all cases, simulations were run assuming mean cross-protections between dengue serotypes of 6 months, 1 year, and 2 years. Outputs had a monthly resolution.

| Sim. | $L$ | $N$ | Temperature input | Aim |
|---|---|---|---|---|
| *1* | 72 | $2.0 \cdot 10^5$ to $5.6 \cdot 10^6$ | Real temperature time series | Reproduce empirical patterns using temperature as the input. |
| *2* | 6 | $1 \cdot 10^6$ | Seasonal sinusoids | Reproduce asynchrony using different thermal regimes alone. |
| *3* | 6 | $1 \cdot 10^6$ | Seasonal sinusoids, with a superimposed single multiannual fluctuation (with a timescale of 2–5 years). | Reproduce synchrony in an asynchronous system and explain contributions of intrinsic and extrinsic factors. |

the multiannual wave was a proportion (10% to 40%) of the seasonal amplitude. The objective of simulation 3 was to determine whether the common multiannual fluctuation across locations would be sufficient to produce synchrony in an otherwise asynchronous system. Using idealised sinusoidal temperature functions in simulations 2 and 3 allows us to better identify whether frequencies detected in the resulting modelled dengue dynamics are due to frequencies present in temperature or due to immunity in the host population.

**Driving the model with actual temperature time series.** The mechanistic model, driven by temperature with the mediation of temporary serotype cross-protection, was capable of producing qualitatively similar dynamics to those observed in the data (e.g., see S12 Fig in S1 Appendix), thereby supporting the hypothesis that temperature plays an important role. The ratios between reported cases and our modelled output (median of 0.11 across all provinces and points in time) are up to almost 30 times greater (for a cross-protection of 1 year) than those estimated in Reich and colleagues [19] (median values of between 0.0037 and 0.0120 across serotypes), suggesting that transmission rates in the model may be comparatively low. However, our results were robust to changes in transmission (see section "Robustness tests" in S1 Appendix). For simulation 1 (using temperature time series for the 72 provinces), the WMF for the model output had statistically significant synchrony at all times (Fig 2C). The Pearson and Spearman correlations between the modelled dengue output and the dengue data WMFs (Fig 2B and 2C) were 0.29 ($P = 0.017$) and 0.32 ($P = 0.011$), respectively. That these correlation coefficients were statistically significant, but those between dengue cases and temperature were not, suggests that the nonlinearities in temperature encoded in the model, together with cross-protection, are important elements of dengue dynamics. These correlation coefficients were still significant for different assumptions on the mean duration of cross-protection (S1 Table in S1 Appendix). The phase angles between temperature and model output across provinces were, as expected, consistent and statistically significant over most time points (Fig 2E). WMFs and CWMFs for model outputs using different assumptions on cross-protection are shown in S13 Fig in S1 Appendix and show that with longer mean cross-protections (e.g., 2 years), the consistency in the phase angles between modelled dengue and temperature is lower than with shorter cross-protections. Both the WMF and CWMF for the model output contain a notable synchrony event in the early 1970s, reproducing that found in temperature; this feature is absent from the WMF and CWMF estimated using the data (Fig 2B and 2D). We also compared the correlations between the wavelet spectra of the modelled dengue output and dengue data for each province, for low- and high-synchrony periods separately. The correlations in high-synchrony conditions were also significantly greater than during low synchrony, but here, low-synchrony correlations were on average greater than the low-synchrony correlations between temperature and dengue data (see S2 Table and S16 Fig in S1 Appendix). These results show that the model (which includes dynamics of immunity) captures mechanisms at play in each province during low synchrony and explains some of the variation in multiannual dynamics of dengue that temperature alone cannot.

**A temperature gradient can generate asynchronous dengue dynamics.** For the idealised experiments of simulation 2, we found that with at least a mean of 1 year temporary cross-protection between serotypes, the resulting dengue dynamics include multiannual cycles (Fig 4A–4C), cycles that are completely attributable to the dynamics of immunity given that here there are no multiannual cycles in temperature. The periodicity of these cycles (and their amplitude), however, changed as a result of the temperature regime. As temperature conditions are optimised for transmission (i.e., as mean temperatures increase, approaching the higher temperatures that characterise temperature in, for instance, Bangkok; S42 Fig in S1 Appendix), the periodicity of multiannual components in dengue go down (i.e., multiannual cycles are more frequent), and the power at the multiannual timescales increases. At the highest mean

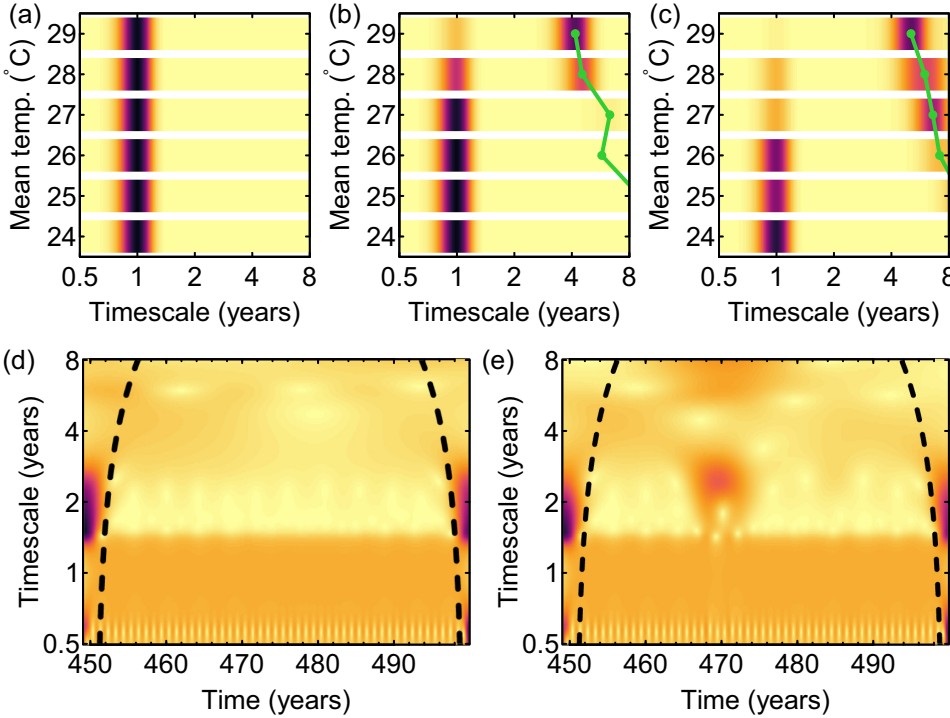

**Fig 4. Multiannual periodicities and synchrony for models driven by simple seasonal sine curves.** (a–c) Mean wavelet power for simulation 2 experiments (seasonal temperature input with no multiannual components; Table 1) per timescale for each of the 6 hypothetical locations (rows within each panel), assuming a mean cross-protection of (a) 6 months, (b) 1 year, and (c) 2 years. (d) Mean wavelet field for the simulations in (b) (i.e., assuming a cross-protection of 1 year) and (e) mean wavelet field for the same simulation, except for the introduction of a single 4-year fluctuation across all locations on year 468. The amplitude of the multiannual fluctuation here is 20% of that of the seasonal cycle. Darker colours indicate greater wavelet power (colours in a–c and d–e are (separately) on the same scale). Green lines in (a–c) indicate peaks in mean wavelet power over multiannual timescales. Edge effects in the WTs may influence results before and after the dashed lines in (d, e). See S19–S32 Figs in S1 Appendix for more complete results (across all cross-protections, and timescales and amplitudes of the multiannual fluctuation). The underlying data are in S1 Data at https://github.com/UF-IDD/synchrony_dengue_figures. WT, wavelet transform.

temperature, with a mean of 2 years of cross-protection, seasonal cycles all but disappear (Fig 4C) due to 2 main trends. On the one hand, as cross-protection becomes longer, the power of the multiannual cycles (and, therefore, their amplitude) increases (Fig 4A–4C). On the other, the seasonal variation in transmission is most limited at the highest mean temperatures (S44 Fig in S1 Appendix), thus producing more limited seasonal variation in dengue cases. Taken together, these 2 trends mean that dynamics become increasingly dominated by the multiannual cycles produced through cross-protection and less so by seasonal variation in temperature. While the periodicity of the multiannual cycles for each temperature regime changes little between means of 1 and 2 years of cross-protection, their prominence does: With longer cross-protection, these cycles increase in mean wavelet power. The range of dynamics produced across locations under all assumptions of cross-protection are sufficiently diverse: No synchrony is detected (e.g., Fig 4D). Nevertheless, part of the asynchrony observed in dengue cases (Fig 2B) may also be due to asynchrony in the multiannual cycles in temperature across locations.

**Common multiannual fluctuations in temperature can synchronise dengue dynamics.** Because different temperature regimes can produce distinct dengue dynamics (Fig 4A–4C), one simple explanation for synchrony could be that temperatures across the country converge

to a more similar temperature regime during those periods, resulting in more synchronous dengue dynamics. However, this has not been the case in Thailand (S38 Fig in S1 Appendix). The next hypothesis is that a common multiannual fluctuation in temperature can synchronise dengue dynamics. The result for simulation 3 was that when the amplitude of the multiannual fluctuation in temperature was at least 20% of the seasonal cycle, the 6 locations were temporarily synchronised (Fig 4E), under all assumptions on cross-protection. Synchrony in dengue was typically detected across a range of timescales. However, when the multiannual fluctuation had an amplitude 10% that of the seasonal cycle and mean cross-protection was 2 years, synchrony was weaker (S32 Fig in S1 Appendix). Synchrony was also generally weaker when the multiannual fluctuation in temperature had a longer periodicity (e.g., 5 years) and a cross-protection of at least 1 year.

**The roles played by immunity and temperature.** The synchronisation achieved in the simulation 3 experiments varied as a function of the mean duration of cross-protection: The degree of synchrony fell with a longer cross-protection (S24–S32 Figs in S1 Appendix). To explain why, we need to better understand how the multiannual timescales produced by intrinsic and extrinsic factors interact. Comparing simulations with (simulation 3) and without (simulation 2) a single multiannual fluctuation in temperature allows us to quantify how the multiannual timescales in dengue change specifically as a result of the multiannual fluctuation in temperature. We found that while the periodicity of the multiannual fluctuation in temperature is generally detectable in the resulting dengue dynamics, its importance relative to the timescales of the intrinsic dynamics (i.e., the dengue multiannual timescales attributable to cross-protection alone) varies as a function of mean temperature, seasonal amplitude in temperature, and mean cross-protection (Fig 5). Dengue appears to be more insensitive to multiannual fluctuations in temperature with increasing mean temperatures, smaller seasonal variation in temperature, and longer cross-protections. Specifically, as the duration of cross-protection and/or mean temperatures increase, and as seasonality in temperature is reduced, the power of the multiannual intrinsic dynamics increases, such that the effect of the multiannual fluctuation in temperature can be overwhelmed. In our simulations, this is particularly the case for higher mean temperatures and a mean cross-protection of 2 years (Fig 5).

It is also worth noting that the multiannual fluctuation in temperature is not necessarily straightforwardly reproduced in dengue dynamics. Instead, it elicits a transient response in dengue cases that can extend in time beyond the multiannual fluctuation in temperature, as can be clearly seen when plotting the phase angle between simulations with a multiannual fluctuation in temperature, and those without (S33 Fig in S1 Appendix). These show both that the effect of the multiannual fluctuation in temperature can be lasting and also confirm that the effect of the multiannual fluctuation in temperature is reduced with longer cross-protections and higher mean temperatures.

This interaction between temperature and cross-protection explains why with longer cross-protection, the degree of synchrony appears to go down: As cross-protection becomes longer, locations with higher mean temperatures are more likely to be dominated by the intrinsic timescales, and, as a result, the ability of a multiannual fluctuation in temperature to synchronise locations is reduced.

## Discussion

Using multiple approaches, we have described an interesting spatial phenomenon in dengue: The country has repeatedly and periodically moved in and out of synchrony in dengue cases. In general, periods of greater synchrony coincided with larger outbreaks. When analysing temperature time series across Thailand, we found a similar pattern in spatial synchrony. Although

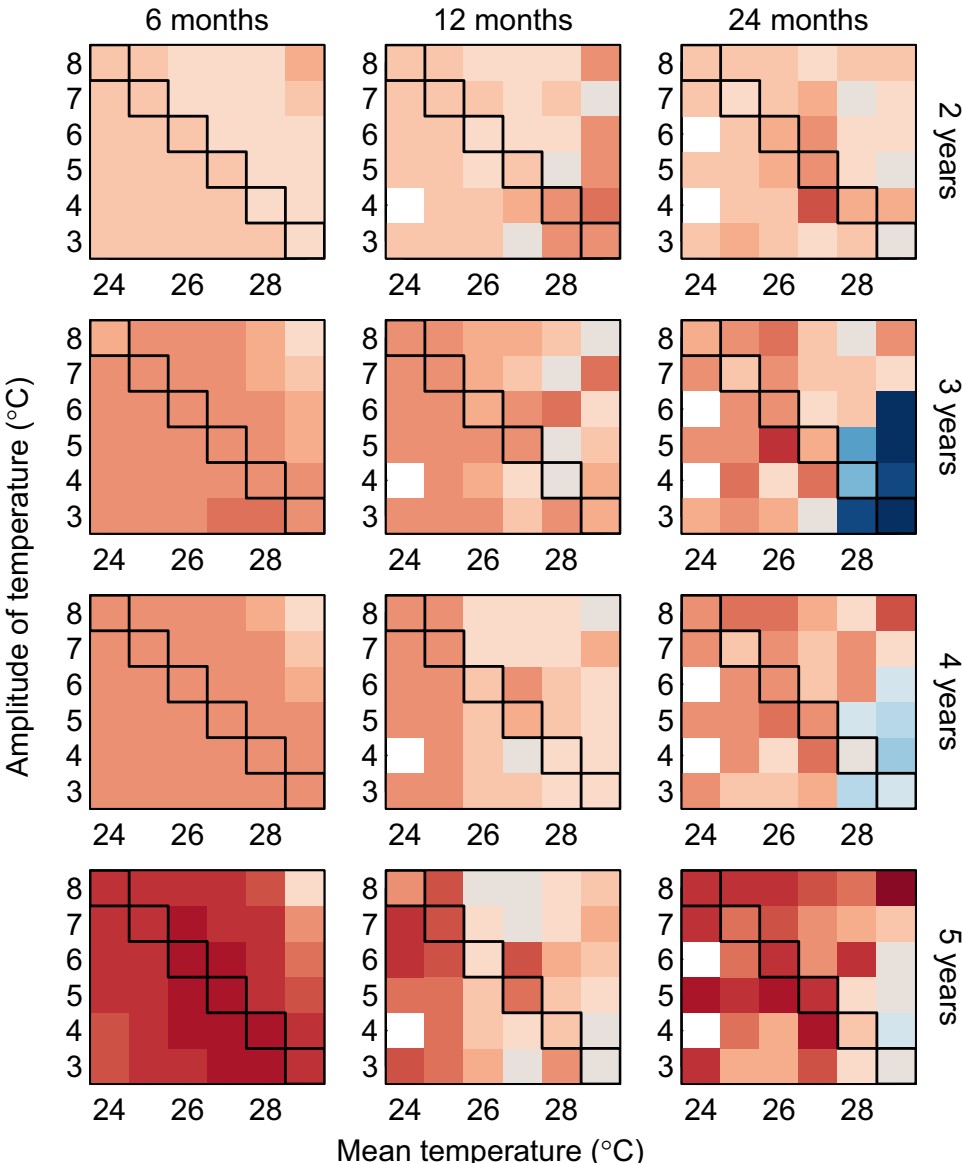

**Fig 5. Comparing multiannual timescales in dengue caused by cross-protection and by multiannual fluctuations in the environment.** We compare the mean wavelet power of dengue at the timescale of the single environmental multiannual fluctuation and at the timescale at which wavelet power is maximised without a multiannual fluctuation in temperature (the multiannual timescale attributable to cross-protection alone) by taking their difference. Columns and rows of panels are for the different assumptions on cross-protection and the different timescales of the multiannual fluctuation in temperature, respectively. Positive values (reds) mean that the multiannual fluctuation in temperature plays a more important role driving dengue multiannual dynamics, and negative values (blues) mean that dengue dynamics are less sensitive to the multiannual fluctuation in temperature. For example, for a mean temperature of 29˚C, a seasonal amplitude of 3˚C, a 3-year fluctuation in temperature, and a mean cross-protection of 2 years, we compared the mean wavelet power at a timescale of 3 years (that of the temperature multiannual fluctuation), with the mean wavelet power at a timescale of 5.1 years (the multiannual timescale at which wavelet power was maximised with no multiannual fluctuation in temperature, i.e., the peak on the top row of Fig 4C). The mean wavelet power was distinctly greater at the 5.1 year timescale than at the 3-year timescale, meaning dengue dynamics were insensitive to the multiannual fluctuation in temperature. Overall, temperature appears to be more important, but longer cross-protections (the right column) and higher mean temperatures are more likely to lead to a system dominated by intrinsic dynamics and less sensitive to multiannual fluctuations in temperature. Mean wavelet power is estimated for the duration of the multiannual fluctuation only (so for a 3-year fluctuation in temperature, mean wavelet power is estimated during those 3 years only). Here, the multiannual fluctuation has an amplitude 0.2 times that of the seasonal cycle, but the patterns are qualitatively similar for the other amplitudes of the multiannual fluctuations. The 6

hypothetical locations are marked along the diagonals. White tiles are missing values due to simulations without a multiannual fluctuation in temperature lacking any peaks in wavelet power at multiannual timescales. The underlying data are in S1 Data at https://github.com/UF-IDD/synchrony_dengue_figures.

the correlation between the overall patterns in synchrony in temperature and dengue cases was not statistically significant, the relationship between temperature and dengue cases was particularly consistent across provinces during times of greater synchrony. Temperature played a less important role during periods of asynchrony. Accounting for the dynamics of immunity is essential for understanding dengue dynamics; any effects of temperature on dynamics are modulated by immunity. To gain a more mechanistic understanding of how temperature interacts with the dynamics of immunity to produce the observed patterns in synchrony, we adapted a mechanistic dengue model with temporary cross-protection. When running the model using the observed temperatures across Thai provinces, the patterns in synchrony in the resultant model output dengue time series correlated positively and significantly with observed patterns in synchrony in dengue cases from the passive surveillance data. The fact that this correlation was statistically significant, but the one between the patterns in synchrony in dengue cases and temperature was not, further highlights the importance of nonlinearities in the way temperature affects dengue and dynamics of immunity. Using the model output, we found that immunity also plays a more important role during periods of asynchrony. While (a sufficiently long) cross-protection alone can produce multiannual cycles, we found that their periodicity was modulated by temperature (which, in turn, drives transmission); different temperature regimes produced characteristically different dengue dynamics. However, although the timescales of the multiannual periodicities of dengue varied little, their power became more prominent with longer cross-protections. We also found that a range of temperature regimes produced asynchronous dengue dynamics, but the introduction of common multiannual fluctuations across a range of different temperature regimes was sufficient to synchronise the otherwise asynchronous system. The degree of synchrony depended on the duration of cross-protection: As the duration of cross-protection increased, the multiannual cycles produced by cross-protection became more prominent compared to those produced by multiannual fluctuations in temperature, particularly at higher mean temperatures, resulting in a lower degree of synchrony across locations. With longer cross-protections, a larger proportion of the population is temporarily protected and not susceptible to infections and thus unaffected by variations in transmission. However, particularly with less than 2 years of mean cross-protection, synchrony in temperature can synchronise an otherwise asynchronous system. Our results highlight how ongoing climate change, together with other concurrent changes, may be affecting dynamics in areas where the disease is endemic and a majority of the population at risk, now and in future projections, lives.

There are several caveats to our analyses. We chose to focus on temperature because we have specific theories for how it might mechanistically impact dengue dynamics. However, we could not ascribe every pattern in synchrony in dengue to synchrony in temperature. For example, Thailand experienced a strongly synchronous 2-year fluctuation in temperature around 1972, and while this fluctuation was reproduced in our model output, it was conspicuously absent from the observed dengue dynamics. There are multiple reasons that could explain this discrepancy. During the earlier part of the records, dengue cases might have been detected less effectively, thus potentially not qualitatively capturing multiannual features of the underlying dynamics. On the other hand, other environmental variables (notably precipitation) and their interaction with temperature are likely important for the dynamics of the vector of dengue. For instance, the extent to which a synchronous event in temperature can synchronise multiannual cycles in dengue might depend on the amount of precipitation or timing of the rainy seasons during those years. Additionally, other nonenvironmental mechanisms might also influence synchrony in dengue. Complex multiannual cycles, synchrony, and variations in the degree of synchrony over time can in

theory be produced via, for example, the movement of hosts between locations [15]. There are also a range of concurrently ongoing changes in Thailand that may also influence multiannual dynamics and synchrony. For instance, the birth rates across the country have been declining [53], and the rates of decline may be spatially heterogeneous. Similarly, long-term trends in population densities across the country could also contribute to the observed spatiotemporal patterns. Exploring how these factors may interact with temperature and contribute to the observed spatiotemporal patterns deserves further attention. The mechanistic model we use here, when compared to data, also leads to reporting ratios that are distinctly higher than those estimated in the literature [19], implying transmission rates in the model may be low. Lower transmission rates might be expected to affect the dynamics of immunity, as susceptibles would be depleted at a slower rate, and would lead to an increase in the periodicity of multiannual cycles produced by immunity. Nonetheless, our results on synchrony are robust to variations in transmission. Finally, the model we use necessarily makes simplifying assumptions (see section "Caveats on the dengue model" in S1 Appendix). Further work is required to understand which of these assumptions may significantly affect synchrony.

The patterns in synchrony in temperature we have observed arise as a result of broader climatological patterns. Specifically, the synchronous events in temperature (Fig 2A) coincide with strong El Niño and La Niña events (1972 to 1973, 1982 to 1983, 1997 to 1998, and 2014 to 2016), during which respectively higher and lower than average temperatures were recorded across the country [54]. The multiannual timescale at which synchrony is detected might then depend on the extent to which El Niño events are followed by La Niña events and their respective strengths. Because of the broad geographical reach of the effects of the El Niño Southern Oscillation (ENSO), we might therefore expect temperature to synchronise dengue dynamics in other countries where ENSO has a similar manifestation in local climate conditions, as shown, for example, in van Panhuis and colleagues [40]. Indeed, several studies have sought links between dengue dynamics and ENSO (e.g., [35,38,40,55,56]). Here, our aim was to test specific hypotheses on how temperature, an environmental variable that we know affects the vector of dengue, can produce asynchrony and synchrony in dengue cases. Care should be taken when drawing the link between ENSO and dengue dynamics, because ENSO affects the environment beyond just temperature, and the physical manifestations of ENSO in any location are complex and can change over time [57]. For example, precipitation, which likely also plays a role in dengue dynamics, can also vary with ENSO (drier conditions associated with El Niño, wetter with La Niña; [58]), but in Thailand, the correlation between precipitation and ENSO has increased over time, while in India, the opposite trend is true [59].

Any disease that is primarily driven by extrinsic (environmental) factors might exhibit a similar pattern of coinciding and related synchronising events in the environment and in the disease, as we observed in this study. The feature that makes dengue a complex system in the context of our study is that intrinsic dynamics can have variation at similar (multiannual) periodicities as the extrinsic drivers (temperature), leading to difficulty in disentangling the role of each in this system. This feature is common to other disease systems, and disentangling extrinsic and intrinsic factors has been a major question in previous work [29–34]. Here, we have applied and developed a series of methods (CWMFs, correlations at low and high synchrony) and used simulations to better understand the relative role of each. Our approach might therefore be useful to other systems where there is a similar intermingling of timescales and for which intrinsic dynamics might constitute an important filter to consider (as might be the case in, e.g., Zika, chikungunya, respiratory syncytial virus, and influenza).

Most locations' temperature appear to be increasing in mean and decreasing in seasonal variation (S39 and S42 Figs in S1 Appendix). If these trends were to be maintained, our results suggest that dengue dynamics could slowly shift, too, and approach a regime characterised by

higher average incidence (S19–S21 Figs in S1 Appendix), and dynamics increasingly dominated by multiannual, rather than seasonal, cycles. Multiannual cycles may also be, to greater extents, dictated by intrinsic factors over the multiannual timescales present in temperature, meaning that in the future, temperature may be less able to synchronise dengue dynamics. However, these hypotheses are contingent on a more precise understanding of the interactions between serotypes and contributions from other extrinsic (e.g., rainfall) and intrinsic factors (such as changes in demography and movement of hosts between locations).

## Materials and methods

### Data

We use data provided by the Thai Ministry of Public Health in their Annual Epidemiological Surveillance Reports. Monthly dengue case counts are given per province, starting in January 1968. Prior to 2003, the counts provided combine the cases of dengue fever (DF), dengue shock syndrome (DSS), and DHF, while separate counts are given for each category thereafter. For 2003 onwards, we use DHF cases only given that these are the cases that are most likely to lead to severe outcomes and the most likely to have been reported prior to 2003, although results change little when including DF and DSS. Starting in 1982, 5 new provinces were created. To maintain consistency across the time series, cases for these new provinces were added back to the provinces they were created from, keeping the total number of provinces (72) constant over time.

Temperature time series were obtained from the GHCN CAMS gridded (0.5˚ by 0.5˚ resolution) monthly mean temperature data set (provided by NOAA/OAR/ESRL PSD, Boulder, Colorado, USA, from https://www.esrl.noaa.gov/psd/data/gridded/data.ghcncams.html, downloaded on 18 June 2019). This dataset combines station observations from the Global Historical Climatology Network v2 and the Climate Anomaly Monitoring System [60] (S37 and S38 Figs in S1 Appendix). The time series start in 1948. We downloaded shapefiles for Thai provinces from https://data.humdata.org/dataset/thailand-administrative-boundaries on 12 February 2019, and estimated province centroids using function "calcCentroid" in R package "PBSmapping" v2.72.1. A time series for each province was then obtained by using the grid point nearest to the centroid of each province.

For simulations using real temperature time series (simulation 1; Table 1), we use the 2020 population sizes of each province (rounded to the nearest $1 \cdot 10^5$) downloaded from the Thai National Statistical Office (http://statbbi.nso.go.th/staticreport/page/sector/en/01.aspx), on 24 November 2021. As with the numbers of cases, we combine populations for provinces created from 1982 onwards.

### Wavelet transforms

We applied continuous WTs to explore how the oscillatory behaviour of dengue cases has changed over time. WTs have now been applied extensively in the study of infectious disease dynamics [37,40,56,61,62] and more broadly in ecology [50,51]. WTs decompose time series into frequency components, but, as opposed to Fourier transforms, WTs are also localised in time. Thus, WTs can be used to characterise nonstationary time series and how the relative importance of different frequencies change over time. The basis for the WT is a "mother" wavelet, a wave localised in both frequency and time (i.e., of limited duration), which is then scaled or stretched (for the frequency component) and shifted along the time axis (for the temporal component) to derive a set of "daughter" wavelets. The WT can then be understood as a correlation between (or more specifically the convolution of) the time series and the set of daughter wavelets [63].

We use the WT as implemented in R package "WaveletComp" v1.1—details are provided in the package documentation [64] and code, and a summary is given here. The package uses a

Morlet mother wavelet, $\psi(t) = \pi^{-1/4} \exp(i \omega t) \exp(-t^2/2)$, where $t$ is time ($t = 1,\ldots,T$) and $\omega$ is the angular frequency, set to 6 radians $t^{-1}$ [64]. The WT, $W$, at a point in time $\tau$ and scale $s$ (proportional to period, the inverse of frequency) is:

$$W(\tau, s) = \sum_t x(t) \frac{1}{\sqrt{s}} \psi^* \left( \frac{t - \tau}{s} \right),$$

(1)

where $x(t)$ is the original time series, and $^*$ denotes the complex conjugate. The modulus of $W(\tau,s)$, $|W(\tau,s)|$, gives the local amplitude $A$ at time point $\tau$ and scale $s$. For any plots of the WT or calculations performed directly on it, we correct $A$ so that $A(\tau,s) = s^{-1/2} |W(\tau,s)|$ [65]. $A(\tau,s)^2$ is the wavelet energy density. $W(\tau,s)$ also yields the instantaneous local wavelet phase. Although the WT contains significant redundancy in time and scale, it is possible to reconstruct the original time series (or time series containing specific scales only) on the basis of summing over the real part of the WT [63].

For these analyses, prior to performing the WT, a value of 1 was added to time series of dengue cases prior to ln-transforming and normalising to a mean of 0 and a standard deviation of 1. Time series of dengue cases were not detrended prior to calculating wavelet power because detrending a time series with zeroes produced artificially odd patterns. Time series of temperature were not transformed; long-term nonlinear trends were removed by taking the residuals of a local polynomial regression (using function "loess" with a span of 0.75), and the detrended time series were subsequently normalised, as above. Due to the finite length of the time series, whenever the wavelets extend beyond the edges of the time series, estimates of transform coefficients become less accurate. This issue is exacerbated as scales increase, due to wavelets extending further in time [63,64]. Regions where these edge effects are present (often referred to as the "cone of influence") are clearly indicated in wavelet plots by thick dashed lines.

While the dynamics of dengue are seasonal, previous studies have also identified distinct multiannual timescales [27,35,37,40,56]. We are specifically interested in these multiannual timescales; for this reason, we focus on timescales greater than 1.5 years. This lower bound prevents leakage from the annual signal into the multiannual timescales. As the timescales increase, edge effects increasingly dominate and the length of time we can reliably analyse becomes shorter. Furthermore, previous studies [35,40] typically identified multiannual components at 2 to 3 years, albeit using shorter time series. For these reasons, we here focus on the 1.5- to 5-year timescales (henceforth, "multiannual" components or timescales refer to this range). See S43 Fig in S1 Appendix for time series of dengue and temperature and their respective WT for 4 representative Thai provinces characterised by different thermal regimes.

## Wavelet mean fields as a measure of synchrony

WMFs provide a measure of synchrony as a function of both scale and point in time (further details can be found in Sheppard and colleagues [50,51]; only a summary is provided here). More intuitively, WMFs indicate scales and time points at which both phases and magnitude of oscillations are consistent (or more synchronous) across provinces. WMFs are a mean of the power-normalised WTs of each location $n$ across all $N$ locations ($n = 1,\ldots,N$), where we normalised each WT by

$$w_n(s, \tau) = \frac{W_n(s, \tau)}{\sqrt{\frac{1}{NT} \sum_{n=1}^{N} \sum_{\tau=1}^{T} W_n(s, \tau) W_n(s, \tau)^*}}$$

(2)

[50,51].

To test whether synchrony is statistically significant in the multiannual range against a null hypothesis of no synchrony, we focus on the consistency of phases across locations (see section "A note on estimating significance of (cross-)wavelet mean fields" in S1 Appendix for an explanation as to why significance of WMFs focusses on consistency of phases). We do so by estimating the wavelet phasor mean field (WPMF), defined as

$$\frac{1}{N}\sum_{n=1}^{N}\frac{W_n(s,\tau)}{|W_n(s,\tau)|},\tag{3}$$

which retains information on the complex phases of the transforms [51]. When different locations have a similar phase, the value WPMF will be large, while when the phase in each location is independent from all other locations, the value of the WPMF will be small. We test significance by generating surrogate datasets where the autocorrelation structure within each time series is preserved. We do this by generating time series with the same Fourier spectrum as the original time series [51]. Following this approach, we produced 1,000 surrogate time series, using function "surrog" in R package "wsyn" v1.0.2 [66]. For each surrogate dataset, we produced a WPMF, and then, for each point in time, and across the multiannual scales, we

1. compared, scale by scale within a single time point, the power of the "real" WPMF with that of all surrogate WPMFs, noted which scales were at least in the 95 percentile, and then calculated the fraction of scales at that point in time for which this was the case (producing a time series of proportions);

2. repeated step 1 but taking each surrogate WPMF in turn and comparing it to all other surrogate WPMFs (producing a time series of proportions for each surrogate WPMF); and, finally,

3. calculated the points in time for which the time series of step 1 were at least in the 95 percentile compared to the time series in step 2.

## Drivers of synchrony

To assess the association in the patterns in synchrony between the WMFs of dengue cases and temperature and between the WMFs of dengue cases and modelled dengue output, we perform both Pearson and Spearman correlations on the respective WMF values within the time-scales of interest for all points in time (excluding the cone of influence). These correlations quantify the overall similarity in the pairs of WMFs and, when statistically significant, would imply that the 2 WMFs are related (and thus lend support to the hypothesis that the patterns in synchrony of temperature drive patterns in dengue). For the Pearson correlations, we square root transform the WMF values. We estimate statistical significance by producing null distributions of correlations. We generate WMFs for 1,000 surrogate datasets of dengue cases, obtained, as above, by preserving the same Fourier spectrum as the original time series of dengue cases. The Pearson and Spearman correlations between each of these surrogate WMFs and the temperature WMF or the model output WMF produce the null distributions against which we compare the observed correlations.

To determine in greater detail whether the patterns of synchrony in temperature were consistent with those observed in the dengue cases, we extended the method above for WMFs [51] to calculate CWMFs. Cross-wavelets quantify the similarity in 2 time series' (temperature and dengue cases) wavelet power at each scale and time point, so the CWMFs quantify the consistency of this similarity across locations. CWMFs are an average of the power-normalised cross-wavelets at each location between temperature and dengue cases. A cross-wavelet $X$

between 2 time series is defined as

$$X(s, \tau) = \frac{1}{s} W_1(s, \tau) W_2(s, \tau)^*, \tag{4}$$

where the cross-wavelet is corrected following Veleda and colleagues [67]. The argument of $X$ gives the phase angles between the 2 time series per time point and scale. For time series of temperature and dengue cases with respective power-normalised (Eq (2)) wavelets $w_n^m(s, \tau)$ and $w_n^d(s, \tau)$, the CWMF is defined as

$$C(s, \tau) = \frac{1}{N} \sum_{n=1}^{N} w_n^d(s, \tau) w_n^m(s, \tau)^*. \tag{5}$$

$C(s,\tau)$ will be large if the phase differences between temperature and dengue for each time point and scale are consistent across locations and if the amplitudes of the oscillations are correlated. Two unrelated but temporally and spatially autocorrelated variables can quite readily produce patterns in synchrony. For example, if instead of using temperatures for Thailand we were to produce a CWMF using temperatures from a different location, with similar temporal and spatial autocorrelation, we could potentially observe similar patterns in synchrony, despite the 2 variables being clearly unrelated. For this reason, we test significance in the multiannual timescales using the same 3-step approach described above, but here, surrogate dengue datasets are produced that preserve not only the autocorrelation structure of the time series, but also the cross-correlation structure across locations (i.e., synchrony-preserving surrogates). We do this by generating time series with the same Fourier spectrum as the original time series (as above), but then adding the same random uniformly distributed phase at each frequency across all time series [51]. As above, we test significance by focussing on the consistency of phase angles between dengue and temperature. Using these surrogate dengue datasets, cross-wavelets are estimated relative to the real temperature dataset, and these, in turn, are used to produce the surrogate cross-WPMFs, which we use for comparison.

Consistency in phase differences and correlations in the amplitudes of temperature and dengue (i.e., higher values of CWMF) might be expected during synchronous periods and would point to the importance of temperature as a driver. On the other hand, low values of CWMF can arise due to multiple reasons that this approach cannot distinguish between. Temperature might also be the driver of dengue dynamics during asynchronous periods, but if multiannual dynamics of temperature were to differ across locations (and, therefore, those of dengue too), both would have low synchrony. On the other hand, dengue dynamics might be primarily driven by intrinsic dynamics, which across locations might produce different multiannual periodicities, thus leading to asynchrony. To address this issue and provide a better explanation for the asynchrony we observe, we performed additional analyses. Using the dengue data WMF values (for multiannual timescales only and excluding the cone of influence), we defined points in time and timescales (henceforth, "conditions") that were above the 75th percentile as "high synchrony" and those below this value as "low synchrony" (S15 Fig in S1 Appendix). Then, for each province, we calculated the Spearman correlation between the square amplitudes of the wavelet spectra of dengue cases and temperature, separately for high- and low-synchrony conditions, yielding 2 correlation coefficients ($r$ values) per province. Were temperature to be the main driver of dengue dynamics during both periods of synchrony and asynchrony, we would expect positive and perhaps similar $r$ values. Systematic differences in $r$ values between high- and low-synchrony conditions would point to the possibility of there being different mechanisms at play. For example, low or no correlations during low-synchrony conditions could be ascribed to the greater importance of intrinsic dynamics or the lack of multiannual cycles in temperature, dengue, or both. We also performed

the same analysis comparing the wavelet spectra of dengue model output and dengue data. In this case, greater $r$ values in low-synchrony conditions than those found between temperature and dengue data would mean that the model captures some of the variation that temperature alone cannot, and would suggest that the nonlinearities in how temperature affects dengue and intrinsic dynamics may be playing a greater role.

## Temperature-dependent dengue model

The model is based on that of Huber and colleagues [46], with the difference that we extended their model to include 4 serotypes with temporary cross-protection. We assume at most 2 infections, after which individuals become immune to all serotypes. Vectors are only ever infected once after which they are immune to all serotypes. The vector compartments (and, therefore, transmission) are fully temperature dependent. The temperature reaction norms are the ones used in Huber and colleagues and Mordecai and colleagues [44,46]; we specifically use parameters corresponding to *Aedes aegypti*.

All vector compartments are indicated with a superscript v. Subscripts indicate the serotype and track infection history. The human compartments include susceptibles to all serotypes ($S_0$) and susceptibles to all serotypes but $i$ ($S_i$; having previously been infected by serotype $i$ and fully susceptible to all other serotypes but $i$), individuals exposed to serotype $i$ ($E_i$), or individuals that have previously been infected by serotype $i$ and currently exposed to serotype $j$ ($E_{ij}$), individuals infected by serotype $i$ ($I_i$) or individuals that have previously been infected by serotypes $i$ and currently infected by serotype $j$ ($I_{ij}$). Individuals that have first been infected by serotype $i$ are then immune to that serotype, but after primary infection, they are temporarily protected against all other serotypes too ($C_i$, where the subscript tracks the primary infection). Their rates of change are

$$\frac{dS_0}{dt} = r - S_0 \sum_{k \in \{1,2,3,4\}} b(T)\gamma(T)\frac{I_k^v}{N} - \mu S_0,$$

$$\frac{dE_i}{dt} = S_0 b(T)\gamma(T)\frac{I_i^v}{N} - (\delta + \mu)E_i,$$

$$\frac{dI_i}{dt} = \delta E_i - (\eta + \mu)I_i,$$

$$\frac{dC_i}{dt} = \eta I_i - (\xi + \mu)C_i,$$

$$\frac{dS_i}{dt} = \xi C_i - S_i \sum_{\substack{j \in \{1,2,3,4\} \\ j \neq i}} b(T)\gamma(T)\frac{I_j^v}{N} - \mu S_i,$$

$$\frac{dE_{ij}}{dt} = S_i b(T)\gamma(T)\frac{I_j^v}{N} - (\delta + \mu)E_{ij},$$

$$\frac{dI_{ij}}{dt} = \delta E_{ij} - (\eta + \mu)I_{ij},$$

$$\frac{dR}{dt} = \eta \sum_{\substack{ij \\ j \neq i}} I_{ij} - \mu R.$$

Here, $r$ is host growth rate (set to correspond to a life span of approximately 77 years), $b$ is the mosquito biting rate, $\gamma$ is the probability of mosquito infectiousness (probability of human infection per bite of an infectious mosquito), $\mu$ is the host mortality rate, $\delta$ is the intrinsic incubation period, $\eta$ is the rate of host recovery from infectiousness, and $\xi$ is the rate of loss of cross-protection.

The rates of change of the vector compartments are

$$\frac{dS^{\mathrm{v}}}{dt} = \frac{l(T)s(T)d(T)}{\mu^{\mathrm{v}}(T)}N^{\mathrm{v}}\left(1 - \frac{N^{\mathrm{v}}}{K(T)}\right) - S^{\mathrm{v}}\sum_{k}b(T)\gamma(T)\frac{I_k}{N} - \mu^{\mathrm{v}}(T)S^{\mathrm{v}},$$

$$\frac{dE_i^{\mathrm{v}}}{dt} = S^{\mathrm{v}}b(T)\delta(T)\frac{I_i}{N} - (\varepsilon(T) + \mu^{\mathrm{v}}(T))E_i^{\mathrm{v}},$$

$$\frac{dI_i^{\mathrm{v}}}{dt} = \varepsilon(T)E_i^{\mathrm{v}} - \mu^{\mathrm{v}}(T)I_i^{\mathrm{v}} + \kappa,$$

where

$$K(T) = \frac{\frac{l(T_0)s(T_0)d(T_0)}{\mu^{\mathrm{v}}(T_0)} - \mu^{\mathrm{v}}(T_0)}{\frac{l(T_0)s(T_0)d(T_0)}{\mu^{\mathrm{v}}(T_0)}}N_m^{\mathrm{v}}\exp\left(\frac{-E_{\mathrm{a}}(T - T_0)^2}{k(T + 273.15)(T_0 + 273.15)}\right), \quad (6)$$

and where $l$ is the number of eggs laid per female, $s$ is the probability of egg-to-adult survival, $d$ is mosquito egg-to-adult development rate, $\varepsilon$ is the virus extrinsic incubation period, and $\delta$ is the probability of infection per bite on an infectious host. Their dependence on temperature is modelled using a Brière or a quadratic function [44]. The term $(l(T)\,s(T)\,d(T))/\mu^{\mathrm{v}}(T)$ quantifies the number of surviving offspring produced in a mosquito lifetime. $N$ and $N^{\mathrm{v}}$ are total number of hosts and vectors, respectively.

The carrying capacity $K$ of the mosquito population is necessary to constrain its population growth. $E_{\mathrm{a}}$ is the activation energy, which defines the temperature dependence of $K$. While Huber and colleagues [46] state that $E_{\mathrm{a}} = 0.5$, the code they provide online has $E_{\mathrm{a}} = 0.05$. We use the latter as the more conservative (less temperature-dependent) option (a value of 0.5 produced strongly temperature dependent carrying capacity, and there is little support in the literature for either value), although our results are robust to changes in this assumption (see sections "Caveats on the dengue model" and "Robustness tests" in S1 Appendix). Finally, $k$ is the Boltzmann–Arrhenius constant, $N_m^{\mathrm{v}}$ is the maximum carrying capacity (capped at some multiple $M$ of the total host population), and $T_0$ is the reference temperature at which carrying capacity $K$ is greatest (here equal to 29°C). Huber and colleagues [46] assumed a maximum ratio $M$ of mosquitoes to humans of 2. Here, we chose a value of 1.5 (i.e, $N_m^{\mathrm{v}} = 1.5\,N$; S44 Fig in S1 Appendix) such that the $R_0$ for mean temperatures typically observed in Thailand were within the approximate range previously reported in the literature of 2 to 7 [68–70]. Our results do not change qualitatively with $M = 1$ (see section "Robustness tests" in S1 Appendix). We constantly introduced a small number of infected mosquitoes, $\kappa$, to mitigate situations where population sizes were reduced to very small numbers. We used $\kappa = 1 \cdot 10^{-5}$ / day / serotype. See section "Caveats on the dengue model" in S1 Appendix for a discussion on some of the assumptions made by this model.

The model was run in R v3.6.2 using the "lsoda" integrator in R package "deSolve" v1.24. In all simulations, the starting conditions were as follows: a total host population size corresponding to that of each province for simulation 1, or $1 \cdot 10^6$ for simulations 2 and 3, in all cases all of which are susceptibles except for 1, 2, 3, and 4 infected individuals allocated to each serotype at random, a host birth and death rate corresponding to a life span of 77 years, and, unless otherwise stated, a mean temporary cross-protection between serotypes of 1 year. Section "Further details on simulation studies" provides more information on how the models were set up.

## Supporting information

**S1 Appendix. Notes and details on methods used, robustness tests using different formulations of model parameters, results employing different approaches to quantify synchrony, and additional results.**
(PDF)

## Acknowledgments

We thank Matt Hitchings and Jessica Metcalf for comments on the manuscript and valuable discussions.

## Disclaimers

Material has been reviewed by the Walter Reed Army Institute of Research. There is no objection to its presentation and/or publication. The opinions or assertions contained herein are the private views of the author and are not to be construed as official or as reflecting true views of the Department of the Army or the Department of Defense.

## Author Contributions

**Conceptualization:** Bernardo García-Carreras, Mary K. Grabowski, Justin Lessler, Derek A. T. Cummings.

**Data curation:** Bernardo García-Carreras, Angkana T. Huang, Sopon Iamsirithaworn, Pawinee Doung-Ngern.

**Formal analysis:** Bernardo García-Carreras, Bingyi Yang, Mary K. Grabowski, Derek A. T. Cummings.

**Funding acquisition:** Derek A. T. Cummings.

**Investigation:** Bernardo García-Carreras.

**Methodology:** Bernardo García-Carreras, Lawrence W. Sheppard, Derek A. T. Cummings.

**Supervision:** Derek A. T. Cummings.

**Visualization:** Bernardo García-Carreras.

**Writing – original draft:** Bernardo García-Carreras, Derek A. T. Cummings.

**Writing – review & editing:** Bernardo García-Carreras, Bingyi Yang, Mary K. Grabowski, Lawrence W. Sheppard, Angkana T. Huang, Henrik Salje, Hannah Eleanor Clapham, Sopon Iamsirithaworn, Pawinee Doung-Ngern, Justin Lessler, Derek A. T. Cummings.

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
