## [Editor Report · Decision Letter 0]

24 Feb 2021

Dear Dr. García-Carreras, 

Thank you for submitting your manuscript entitled "Periodic synchronization of dengue epidemics in Thailand: the roles played by temperature and immunity" for consideration as a Research Article by PLOS Biology.

Your manuscript has now been evaluated by the PLOS Biology editorial staff, as well as by an academic editor with relevant expertise, and I am writing to let you know that we would like to send your submission out for external peer review.

Please re-submit your manuscript within two working days, i.e. by Feb 26 2021 11:59PM.

Kind regards,

Paula

---

Associate Editor

PLOS Biology

---

## [Decision Letter · Decision Letter 1]

16 Apr 2021

Dear Dr. García-Carreras,

Thank you very much for submitting your manuscript "Periodic synchronization of dengue epidemics in Thailand: the roles played by temperature and immunity" for consideration as a Research Article at PLOS Biology. Your manuscript has been evaluated by the PLOS Biology editors, an Academic Editor with relevant expertise, and by several independent reviewers.

In light of the reviews (below), we will not be able to accept the current version of the manuscript, but we would welcome re-submission of a much-revised version that takes into account the reviewers' comments. We cannot make any decision about publication until we have seen the revised manuscript and your response to the reviewers' comments. Your revised manuscript is also likely to be sent for further evaluation by the reviewers.

In particular, reviewer #1 says that there are substantial issues with the logic and results of the analyses, and that the evidence for a role of immunity, through cross-protection, is not convincing and remains confusing in its argument, adding that the study in its current form is incomplete. This reviewer thinks that the manuscript needs a better comparison of the modelling exercise to the data, and a clearer reasoning on the role of cross-protection, says that the argument about the role of intrinsic dynamics vs. extrinsic factors is particularly confusing, that you need to compare the simulations to the empirical patterns, and says that the patterns you see are suggestive of a role of El Niño events in the synchrony patterns and that it is incomplete to leave out of an explanation for the synchrony how ENSO is manifested in the regional climate and how this manifestation has changed across events. Reviewer #1 also says that there are inconsistencies between patterns of synchrony in cases and temperature, and between those in observations and simulations, that there are some confusing statements about multiannual cycles, and that they should clarify whether there was a reporting mistake in the Huber et al. paper and why such a different value is required here. Reviewer #2 finds the discussion short and underwhelming, and has suggestions on how to expand it, thinks that you should define additional terms throughout the manuscript to help readability, and add more details on the model in the main text, and, similarly to reviewer #1, how your model differs from that of Huber et al.

We expect to receive your revised manuscript within 3 months. 

**IMPORTANT - SUBMITTING YOUR REVISION**

*Re-submission Checklist*

*Published Peer Review*

*PLOS Data Policy*

*Blot and Gel Data Policy*

Sincerely,

Paula

---

Associate Editor,

pjaureguionieva@plos.org,

PLOS Biology

REVIEWS:

Reviewer #1: Infectious disease dynamics.

Reviewer #2: Quantitative ecology.

Reviewer #1: Major comments:

In this work the authors examine the synchrony of reported cases of dengue (for Dengue Hemorrhagic Fever, or DHF) across Thailand with and unprecedented data set spanning 51 years across 72 provinces. The synchrony of temperature and that of temperature with dengue cases are also addressed. A process-based model for the population dynamics of dengue driven by temperature is used to examine the role in the generation of multiannual cycles of extrinsic forcing (by temperature) vs. that of intrinsic dynamics (through cross-protection between different serotypes). 

The appeal of this study is the spatio-temporal data set, which provides a longer window of time to address the role of climate vs. immunity in the population dynamics of endemic dengue across this large region. Previous studies had already examined a role of climate drivers and synchrony, in particular with reference to the El Niño Southern Oscillation (ENSO). Here, a longer perspective becomes possible. Indeed, the patterns of Figure 1 are interesting.

Unfortunately, there are substantial issues with the logic and results of the analyses. Results do show that dengue in Thailand moves in an out of synchrony, with periods of widespread and significant synchrony. The evidence for a role of temperature on these patterns is fairly convincing, except for some inconsistencies raised below. The evidence for a role of immunity, through cross-protection, is not convincing and remains confusing in its argument. 

My major comments below explain why. They also indicate my concerns with the evidence developed to explain the patterns of synchrony. The study in its current form is incomplete.

The statistical and modeling machinery exists today to address different hypotheses on the synchrony, by fitting to these time series process-based models and applying model selection. I am not asking the authors to go that way, but a better comparison of the modeling exercise to the data, and a clearer reasoning on the role of cross-protection Is needed.

Major comments:

1) Role of intrinsic dynamics vs. extrinsic factors: 

The argument is particularly confusing. The simulations (number 3) include multiannual variation in the synthetic "temperature" forcing (basically a sinusoidal modulated in its amplitude), and then compare the mean wavelet power of the response at the temporal scale of the forcing to the mean wavelet power at the dominant temporal scale of what is considered the "intrinsic" dynamics. The latter is obtained from the same model forced by the seasonal synthetic temperature with no multiannual variation. In other words, the comparison is for the intensity of the multiannual response with and without the multiannual variation in the synthetic temperature (and at different time scales since the dominant multiannual periods differ for those two runs). We are shown that the power of the multiannual variation caused by the "intrinsic" cycles increases for longer duration of cross-protection and higher mean temperature (with lower variance). Why does this imply that immunity is needed to explain the observed patterns of synchrony? The logic is unclear and possibly flawed. We expect higher mean temperatures (if they cause higher transmission intensity and are closer to the optimum for transmission) to make the role of immunity more prominent as the nonlinear dynamics of the disease becomes more important. This does not tell us that the multiannual variation and its patterns of synchrony in the observed cases necessitate a role of cross-protection. The characteristic multiannual time scale generated in this way differs from that in the temperature forcing. Is this the case in the data? Moreover, since the variability of temperature involves multiple time scales (in particular, seasonal and multiannual), it would be important to provide a convincing argument for why it is not a sufficient explanation for the patterns observed in the data. That intrinsic dynamics (just with seasonal forcing) produces under the longest cross-protection considerable interannual variability with a characteristic period shorter than 4 years is not per se evidence that such cross-protection is important in the real system. Comparisons of these simulations to the empirical patterns are needed, rather than between simulations themselves. I would have liked to see the actual patterns of variation in dengue cases as a function of latitude, for representative provinces of the temperature "regime". Do the predictions of Figure 4 c better explain the data than those of Figure 4b? Do we see periods in the dengue dynamics that are not present in the temperature? The power at a given multiannual frequency is not the only aspect of the synchrony that matters; when in time those cycles occur is also important and highly non-random here (see 2 below). One may get multiannual cycles out of "intrinsic" dynamics , even cycles with periods observed in the data, but their timing will not be synchronous across the region without multiannual variation that is extrinsic setting the timing. (On a more technical note, I could not tell whether in the comparisons of Figure 5 the multiannual variation in the synthetic temperature was introduced for a single such cycle like in Figure 4e). 

2) Climate variability as an explanation for the patterns:

By eye, the patterns of Figure 1 are very suggestive of a role of El Niño events in the synchrony patterns. The years when cases are high across Thailand have a high correspondence to some major EN events, namely 1986-87 and 1997-1998. This is consistent with previous studies of dengue dynamics in the region although those studies were more limited in the data. I appreciate the consideration of regional climate (here, temperature) but it seems surprising and incomplete to leave out of an explanation for the synchrony how ENSO is manifested in the regional climate and how this manifestation has changed across events. ENSO is a major driver of the regional climate in Thailand and the way it affects temperature (and also rainfall) should be relevant to the observed patterns, especially when acting on top of the long-term warming trend. 

3) Inconsistencies between patterns of synchrony in cases and temperature, and between those in observations and simulations:

There are some convincing similarities between patterns of synchrony in temperature and cases. There are some exceptions to this correspondence that should be explained in some way. Namely, the synchronous event in early 1970s which is present in the temperature but not in the cases. Because it is present in the temperature, it also shows up in the simulated cases (Figure 2 c). Again, this differs from what is observed in the data. Similarly, the cross-wavelet mean fields in panels (d) and (e) differ for the simulations and the data. Could there be an effect of initial conditions? Of reporting rates in that first part of the data? Of the long-term trend in temperature? Another inconsistency is the non-significant correlation between the wavelet synchrony of dengue and that of temperature, compared to the significant correlation of the same quantities for the simulations. Lines 303-306 argue that this discrepancy highlights the importance of nonlinearities. I do not follow the argument: the nonlinearities should affect both model and observations. 

4) Better understanding of intrinsic dynamics (with just seasonal forcing);

In the population dynamics of infectious diseases, the multiannual variation that is called intrinsic arises from seasonality interacting with the natural (damped or persistent) oscillations caused by the nonlinear dynamics of the disease itself. Here, there are some confusing statements about multiannual cycles arising in part because seasonality itself becomes less important. This is not a given. Seasonality is an integral component of generating the interannual variability. Whether or not weaker seasonality is required we cannot tell, as the mean of temperature grows at the same time that seasonality weakens. Generalizing from such limited simulations seems incomplete. For warmer temperatures and higher transmission, the depletion of susceptibles and effects of cross-protection should also grow.

5) The parameterization of the carrying capacity was difficult to follow, in terms of the value reported in Huber et al. and the one used here. It should be possible to clarify with the authors whether there was a reporting mistake in that paper and why such a different value is required here. The carrying capacity is a very challenging aspect of dengue models where the dependency on rainfall should show up. It is difficult to follow that part of the model and even harder to understand the need for such a different parameter. It would be important to clarify this so that the confusion on this matter does not propagate in the literature.

The following comments are not major ones; they are needed to clarify the work.

6) Why was the trend removed for the cases but not temperature?

7) The statement that endemic regions may be those most strongly affected by climate change is not that meaningful without the context of the range of the reproductive number of the disease and its dependence on temperature. The effect of climate will depend on where a region lies in this dependence. This is so for the same reason that the model used here shows a weaker sensitivity to temperature forcing where the function of the transmission rate with temperature is closer to the optimum. 

8) It would be useful to provide the R0 (or vectorial capacity of the model) as a function of the relevant range of temperatures. The figure of the biting rate and infectiousness with temperature does not consider all the parameters that depend on temperature and affect R0 in the model. 

9) Why use a sinusoidal for the synthetic temperature and not just the seasonality of empirical temperature in a given location? (By seasonality, I mean here the mean temperature pattern as a function of time of the year). It is possible that a sinusoidal fits this pattern well, in which case such a fit could be used. 

10) I would have liked to see representative plots of the time series of cases across the region, especially for those locations that are chosen for the different temperature regimes. Similarly, it would be informative to see the wavelet spectra of these time series. 

11) Please provide some information on the spatial resolution of the temperature data compared to the resolution of the dengue data set. 

12) Figure 2: I would have liked to see significance on the wavelet mean-field plots, although I was unclear on whether this can be done, as the significance appears to be evaluated with a different quantity that concerns the phases. 

13) What is shown in Figure 3 (a)? The caption was unclear about this.

14) I was not convinced by the value of examining the periodicity of the synchrony. What does this tell us given the low number of synchrony cycles, and given patterns dominated by two main events of large synchrony (corresponding to particular El Niño events about 12 years apart)?. 

15) Line 140: clarify how the normalization is performed

16) The abstract would benefit from a complete rewrite of the part that states the findings. 

17) In general, I found the definitions of the different quantities related to wavelets somewhat difficult to follow, especially in places where significance is tested on one quantity but synchrony is evaluated with another. I know the references are provided for the methods. It would be useful anyway to have a more self-contained set of definitions that a general reader can follow here.

Reviewer #2: The authors use a long-term spatiotemporal data set of dengue incidence in Thailand to investigate synchrony in outbreaks across provinces through time. They find that there have been multiple synchronization events, and then use a mechanistic model to illustrate how this pattern can be driven by both temperature and immune dynamics. 

I think this is an exciting and thorough study on a topic that should be of wide interest. I also agree with the authors that studying how climate affects infectious disease in endemic locations is relatively understudied compared to the research focus on disease range expansions and shifts. I have three "major" comments on potential ways to improve the manuscript for a wider audience such as PLOS Biology's.

Major Comments:

1. After the relatively long Results section, I found the Discussion to be short and a bit underwhelming. Approximately half of the Discussion is in the form of a long first paragraph which only really recaps the Results section, and most of the remaining Discussion is dedicated to the final Caveats paragraph. Overall, this doesn't leave much room for more in-depth discussion. 

There are a few possible findings and ideas that could be expanded on in new Discussion paragraphs. I'm not suggesting the authors necessarily need to address all these topics, but I do think the study would be improved with a more thorough discussion of the study's findings and implications for other systems. For example:

* How generalizable should we expect these findings to be for dengue in other countries besides Thailand? Is there anything about Thailand in particular that makes the authors think it should or shouldn't be generalizable to other countries? Has a similar analysis ever been conducted for dengue in other locations? 

* How generalizable should these dengue findings be to other infectious diseases? What about other vector-borne diseases that we know are affected by temperature, like malaria? Has a similar analysis ever been conducted for another infectious disease, and if so, what did they find? 

* Precipitation is mentioned near the end of the Discussion in relation to what other factors besides temperature may have already affected the patterns of synchrony in the data set. However, in addition to temperature, precipitation is also expected to vary with climate change in the future. Are there any reasons to suspect that this will affect predicted future patterns of synchrony?

2. Not all readers of this study will have familiarity with time series synchronization analyses. Of course, some knowledge needs to be assumed, but I think defining a few additional terms throughout the manuscript would still help readability. I'll note that I actually think the authors already do a pretty good job of this; for example, when simply describing what WTs are in the first paragraph of the Results, and later what CWMFs are. However, I think there are a few more places additional text/explanation could be included. For example, defining in plain words terms such as the Moran effect, phase angle, wavelet power, and common multiannual fluctuation.

3. As someone who is interested in the mechanistic modeling, I was hoping to see more details on the model in the main text. Indeed, even though lines 162-163 in the Results say to "see Materials and Methods" for the model, there doesn't appear to be any information or subheading regarding the model in the Materials and Methods. I realize a lot of the details are in the supplementary information, but if the authors are not constrained by space I think it would be beneficial to summarize some of that information and potentially the equations themselves under a new subheading in the Materials and Methods section of the main text. I think it would also be useful to explicitly state in the main text how the authors model differs from that of Huber et al., rather than just stating that it builds on it.

Minor Comments:

* Line 28: unclear what "predator-prey dynamics between the virus and the host" means. Is this referring to dynamics between the virus and vector? And what type of dynamics?

* 325-327: This one sentence paragraph seems a bit out of place.

* Lines 346-347 note that "During the earlier part of the records, dengue cases might have been detected less effectively, thus potentially obscuring multiannual patterns." Are changes in case detection relevant at any other time in the rest of the time series?

---

## [Decision Letter · Decision Letter 2]

11 Oct 2021

Dear Dr. García-Carreras,

Thank you for submitting a revised version of your manuscript "Periodic synchronization of dengue epidemics in Thailand: the roles played by temperature and immunity" for consideration as a Research Article at PLOS Biology. This revised version of your manuscript has been evaluated by the PLOS Biology editors, the Academic Editor and one of the original reviewers.

In light of the reviews (below), we are pleased to offer you the opportunity to address the remaining points from the reviewer in a revised version that we anticipate should not take you very long. In particular, reviewer #1 has an important issue regarding figure 12, saying that the simulated cases should be within the plausible range expected for reporting rates in the region. We will then assess your revised manuscript and your response to the reviewers' comments and we may consult the reviewers again. Please also address the following policy and editorial requests.

DATA POLICY:

2) Deposition in a publicly available repository. **Please also provide the accession code or a reviewer link so that we may view your data before publication.**

Regardless of the method selected, please ensure that you provide the individual numerical values that underlie the summary data displayed in the following figure panels as they are essential for readers to assess your analysis and to reproduce it: Figures 1AB, 2ABCDE, 3AB, 4ABCDE, 5, S1ABCDEF, S2ABC, S3ABCD, S4AB, S5ABCD, S6AB, S7, S8, S9AB, S10ABCDE, S11ABC, S12ABCDEF, S13ABCD, S14, S15AB, S16ABC, S17ABC, S18ABC, S19AB, S20AB, S21AB, S22, S23, S24ABCD, S25ABCD, S26ABCD, S27ABCD, S28ABCD, S29ABCD, S30ABCD, S31ABCD, S32ABCD, S33, S34ABC, S35AB, S36, S37AB, S38AB, S39AB, S40, S41, S42AB, S43, S44.

**Please also ensure that figure legends in your manuscript include information on where the underlying data can be found, and ensure your supplemental data file/s has a legend.**

**Please ensure that your Data Statement in the submission system accurately describes where your data can be found.**

Please provide a blurb which (if accepted) will be included in our weekly and monthly Electronic Table of Contents, sent out to readers of PLOS Biology, and may be used to promote your article in social media. The blurb should be about 30-40 words long and is subject to editorial changes. It should, without exaggeration, entice people to read your manuscript. It should not be redundant with the title and should not contain acronyms or abbreviations. 

We also suggest a change in the title to stress the longitudinal nature of the study, and give a sense of what are the roles played by temperature and immunity on the synchronization of epidemics. We suggests "A more than 50-year study identifies crucial roles for temperature and immunity in the periodic synchronization of dengue epidemics under a global warming scenario". But feel free to modify further, ideally adding the roles played by temperature and immunity.

We also think that the abstract is not very accessible. We suggests you offer it to read to someone outside your field and rewrite it to make it broadly accessible, clearly explaining the role played by temperature and by immunity and why this is important.

We expect to receive your revised manuscript within 1 month.

**IMPORTANT - SUBMITTING YOUR REVISION**

*Resubmission Checklist*

*Published Peer Review*

*PLOS Data Policy*

*Blot and Gel Data Policy*

Sincerely,

Paula

---

Paula Jauregui, PhD

Associate Editor

PLOS Biology

REVIEWS:

Reviewer #1: The paper presents an analysis of the temporal patterns of synchronization of seasonal endemic dengue across space in Thailand. The study is unique in the length of the data set analyzed (5 decades) and the number of spatial locations (72 provinces). The described patterns of localized synchrony with a few episodes of high synchrony/high incidence interspersed among intervals of asynchrony are an interesting finding. The authors also investigate the role of local temperature vs. temperature interacting with immunity, in establishing these patterns. This is done with a process-based model of dengue and its responses to three kinds of forcing: (1) observed temperature, (2) seasonality along a gradient of mean temperatures representative of Thailand (with a pure sinusoidal function), and (3) a single multiannual cycle overlayed on the seasonality. These simulations suggest a role of immunity and its interaction with the extrinsic driver, in particular for the parameters that characterize the more endemic locations. This is not surprising as places with the higher temperatures and therefore, more endemic dengue dynamics, would be the ones where susceptible depletion through patterns of cross-protection play a more important role. Nevertheless, the simulations are valuable to identify parameter ranges/conditions for which this role in interannual variability and therefore, synchrony, would come into play.

In this revision, the authors have addressed my previous comments and the text is now clear. I have one additional major comment arising from the new figures presented in the Supplement. Although this is only one major concern, it is an important one for the relevance of the simulation results and therefore for the conclusions reached with the model.

Major comment:

Figure S12 now shows the actual time series of reported cases together with the simulated cases produced by the model. If I am not mistaken, comparison of the two temporal trajectories shows a difference of about an order of magnitude in the peaks and temporal mean, with the simulations producing lower infections than those observed in the data. A difference would be expected but in the opposite direction, due to under-reporting. Another way to put it is that the model produces levels of infection that are substantially lower than what is consistent with observations, which is an indication that the parameters of the model are possibly problematic and not representative of the dynamics in this region. I understand and value the effort to use parameters from the literature based on metabolic theory but I wonder to what extent these results indicate that this bottom-up approach, with no calibration at all of the model to the data, cannot really account for the transmission dynamics in the region.

A substantial number of papers has taken a different approach to this question of intrinsic vs. extrinsic factors in the population dynamics of climate-sensitive diseases including vector-borne ones. This approach has been based on methods of parameter inference and model selection, with models fitted to time series. (It may be good to cite at least some of this work as a broader context for the questions here including diseases beyond dengue. See for example, Koelle et al. Nature 2005, or Laneri et al. Plos Comp. Biology 2010, and the many papers that followed).

I understand that the approach taken here is different and by design does not involve fitting the model to the data, as such fitting would be challenging for panel data and a disease like dengue with multiple serotypes. Nevertheless, the simulated cases should be within the plausible range expected for reporting rates in the region, and not as far from the data as what Fig. 12 seems to indicate. This is important for the central argument of the paper because incidence reflects the intensity of transmission, and this intensity should in turn be related to how much of a role immune protection plays in generating multiannual variation. This issue deserves at least mention and discussion, if not the re-parameterization of the model, unless I am missing something, or the labels of these time series were exchanged in the figure.

Minor comments:

1) Abstract: in the third line, the text refers to "climate change". Please change to climate variability and change, as this paper is not primarily about climate change per se.

2) Abstract: The sentence "Observed patterns in dengue were statistically more similar…", is somewhat misleading. What is more similar is the pattern of synchrony. (And here comes the problem above, as a reader may interpret this to mean that the model produces patterns of dengue similar to those in the data. For the standards of this kind of work today, this text can give the impression that the models were fitted). Similarly, the sentence "Applying a temperature-driven model…", could be changed to something like "Simulations of a temperature driven model" .

3) Lines 111-119: No mention is made here of the difference in the simulations and data concerning whether synchrony is present/absent in early 1970s in Figure 2 . This difference is discussed much later but not acknowledged where first observed.

4) Figures S19-21: adding the scale of the cases in the y-axis would be informative, especially given the possible discrepancy with the observed cases.

5) Line 335: the correlation is with the patterns of synchrony and not with the observed dengue cases.

6) Lines 424-426: here the text touches on the important implications of the work for what one may expect under climate change. It would seem a subject worth expanding on. I did expect that as more places become endemic, more asynchrony rather than synchrony would be expected.

7) Lines 409-412: This statement about dengue is what led to work in the last twenty years on disentangling extrinsic and intrinsic factors in a number of climate-sensitive diseases. Dengue presents the challenge of multiple co-circulating serotypes but it is not unique in this sense, and this question has been a major one in the literature for some time. Worth writing in a way that acknowledges this.

---

## [Editor Report · Decision Letter 3]

22 Nov 2021

Dear Dr. García-Carreras,

Thank you for submitting your revised Research Article entitled "Periodic synchronization of dengue epidemics in Thailand between 1968 and 2018: the roles played by temperature and immunity" for publication in PLOS Biology. I have now discussed your revision with the Academic Editor. 

We will probably accept this manuscript for publication, provided you satisfactorily address the remaining issues and address the following data and other policy-related requests.

Discussing with the Academic Editor, we think that you need to respond more deeply to the serious criticisms raised by the referee and discussed on page 2 and 3 of the response. The changes made to the manuscript for this response are quite superficial and do not reflect the more than order of magnitude difference between the simulations and the observed data. You should expand your discussion of this in both the figure legend and the accompanying text.

DATA POLICY:

2) Deposition in a publicly available repository. 

Regardless of the method selected, please ensure that you provide the individual numerical values that underlie the summary data displayed in the following figure panels as they are essential for readers to assess your analysis and to reproduce it: Figures 1AB, 2ABCDE, 3AB, 4ABCDE, 5, S1ABCDEF, S2ABC, S3ABCD, S4AB, S5ABCD, S6AB, S7, S8, S9AB, S10ABCDE, S11ABC, S12ABCDEF, S13ABCD, S14, S15AB, S16ABC, S17ABC, S18ABC, S19AB, S20AB, S21AB, S22, S23, S24ABCD, S25ABCD, S26ABCD, S27ABCD, S28ABCD, S29ABCD, S30ABCD, S31ABCD, S32ABCD, S33, S34ABC, S35AB, S36, S37AB, S38AB, S39AB, S40, S41, S42AB, S43, S44.

We realised that you added to the figure legends in your manuscript information on where the underlying data can be found, however, there is no file with the data. Please ensure that your Data Statement in the submission system accurately describes where your data can be found. Please also provide the accession code or a reviewer link so that we may view your data before publication.

Please remove the blurb from the manuscript, as this is only necessary in the our submission system. Thank you for providing one. 

We also suggest a change in the title to stress the longitudinal nature of the study, and give a sense of what are the roles played by temperature and immunity on the synchronization of epidemics. 

We expect to receive your revised manuscript within two weeks. 

*Published Peer Review History*

*Early Version*

Sincerely,

Paula 

---

Associate Editor,

pjaureguionieva@plos.org,

PLOS Biology

---

## [Decision Letter · Decision Letter 4]

7 Jan 2022

Dear Dr García-Carreras,

Thank you for submitting your revised Research Article entitled "Periodic synchronization of dengue epidemics in Thailand between 1968 and 2018: the roles played by temperature and immunity" for publication in PLOS Biology. I have now obtained advice from one of the original reviewers and have discussed their comments with the Academic Editor. 

I should clarify that my colleague Dr Paula Jauregui is now on maternity leave, so I'm handling your manuscript through its final stages. The Academic Editor had asked us to consult reviewer #1 one last time; this reviewer now has one last request, so we are nearly there...

IMPORTANT:

a) Please address the final request from reviewer #1, to clarify this important discrepancy.

b) Please could you improve the title? We suggest the following options: "A 50-year study identifies periodic synchronization of dengue epidemics in Thailand driven/shaped/regulated [as appropriate] by temperature and immunity" or "A 50-year study identifies key roles of temperature and immunity in the periodic synchronization of dengue epidemics in Thailand" or "Periodic synchronization of dengue epidemics in Thailand over the last five decades driven/shaped/regulated by temperature and immunity."

Based on the reviews, we will probably accept this manuscript for publication, provided you satisfactorily address the remaining points raised by the reviewers. Please also make sure to address the following data and other policy-related requests.

***Insert Editorial Requirements HERE, including REPORTING, ETHICS & DATA REQUIREMENTS AS NECESSARY***

We expect to receive your revised manuscript within two weeks. 

*Published Peer Review History*

*Early Version*

Sincerely,

Roli Roberts

Roland G Roberts PhD

Senior Editor

PLOS Biology

rroberts@plos.org

DATA NOT SHOWN?

REVIEWER'S COMMENTS:

Reviewer #1: 

All my comments have been addressed. In particular, the clarification on population sizes and the changes made concerning population size in the model simulations are useful. I agree with the authors that given the functional form of the transmission term in the equations, the general conclusions and patterns of synchrony should not change because the overall dynamics are independent from population size. Still, the actual number of cases is affected, and seeing these numbers is useful for a comparison to the observed values. This comparison matters because it indicates whether the parameters used in the simulations are consistent with the observed cases. This in turn matters because the parameters determine the importance of immunity (ie the depletion of susceptible individuals), and given that the conclusions concern the role of population immunity, I believe it matters to know whether the parameterization is meaningful for this region. I hope this argument clarifies my reason for requesting the further explanation.

One further request: The difference between the order of magnitude of the observed cases and those simulated should be acknowledged in the Results and discussed in the Discussion, in relation to what is known about the reporting rates in the region and the possible implications for the result (on the importance of immunity). I am not requesting that the model be calibrated or fitted to the data but simply that the difference be recognized and discussed. Ultimately, there is the question of whether models parameterized completely from literature values and published curves for the dependence on temperature are able to represent the dynamics of the disease. This is probably not the case for many reasons. It may not matter for the synchrony patterns but it may influence the interpretation of the results. Now the simulated cases are larger than the observed but by one or two orders of magnitude. Is this an indication of too high a transmission rate? Or is it consistent with reporting rates? The reader would want to know.

The paper is a valuable and interesting study. It would be a shame to have this point "hidden" from discussion.

---

## [Editor Report · Decision Letter 5]

24 Feb 2022

Dear Dr García-Carreras,

On behalf of my colleagues and the Academic Editor, Andy Dobson, I'm pleased to say that we can in principle accept your Research Article "Periodic synchronization of dengue epidemics in Thailand over the last five decades driven by temperature and immunity" for publication in PLOS Biology, provided you address any remaining formatting and reporting issues. These will be detailed in an email that will follow this letter and that you will usually receive within 2-3 business days, during which time no action is required from you. Please note that we will not be able to formally accept your manuscript and schedule it for publication until you have any requested changes.

I'd also like to take this opportunity to apologise for an administrative error which has led to a delay at this final stage - when you re-submitted your manuscript it was erroneously assigned to my colleague Dr Paula Jauregui, who is on maternity leave, instead of to me. I only noticed this error today, and re-assigned it to me and assessed your revisions as quickly as I could. So I'm sorry for this delay, and assure you that we will re-examine our processes to ensure that this problem does not recur.

PRESS: We frequently collaborate with press offices. If your institution or institutions have a press office, please notify them about your upcoming paper at this point, to enable them to help maximise its impact. If the press office is planning to promote your findings, we would be grateful if they could coordinate with biologypress@plos.org. If you have not yet opted out of the early version process, we ask that you notify us immediately of any press plans so that we may do so on your behalf.

Sincerely,

Roli Roberts 

Roland G Roberts, PhD 

Senior Editor 

PLOS Biology

rroberts@plos.org